# Assessment of spatial uncertainty of heavy rainfall at catchment scale using a dense gauge network

Sungmin O[1,2,*] and Ulrich Foelsche[1,2,3]

[1]Institute for Geophysics, Astrophysics, and Meteorology/Institute of Physics (IGAM/IP), NAWI Graz, University of Graz, Austria
[2]FWF-DK Climate Change, University of Graz, Austria
[3]Wegener Center for Climate and Global Change (WEGC), University of Graz, Austria
[*]Now at Biogeochemical Integration, Max Planck Institute for Biogeochemistry, Jena, Germany

**Correspondence:** Sungmin O (sungmin.o@uni-graz.at)

**Abstract.** Hydrology and remote-sensing communities have made use of dense rain-gauge networks for studying rainfall uncertainty and variability. However, in most regions, these dense networks are only available at small spatial scales (e.g., within remote-sensing subpixel areas) and over short periods of time. Just a few studies have applied a similar approach, i.e., employing dense gauge networks to catchment-scale areas, which limits the verification of their results in other regions. Using 10-year rainfall measurements from a network of 150 rain gauges, WegenerNet (WEGN), we assess the spatial uncertainty in observed heavy rainfall events. The WEGN network is located in southeastern Austria over an area of 20 km × 15 km with moderate orography. First, the spatial variability of rainfall in the region was characterised using a correlogram at daily and sub-daily scales. Differences in the spatial structure of rainfall events between warm and cold seasons are apparent and we selected heavy rainfall events, the upper 10% of wettest days during the warm season, for further analyses because of their high potential for causing hazards. Secondly, we investigated the uncertainty in estimating mean areal rainfall arising from a limited gauge density. The average number of gauges required to obtain areal rainfall with errors less than a certain threshold (<=20% normalized root-mean-square error is considered here) tends to increase roughly following a power law as the time scale decreases, while the errors can be significantly reduced by establishing regularly distributed gauges. Lastly, the impact of spatial aggregation on extreme rainfall was examined, using gridded rainfall data with various horizontal grid spacings. The spatial scale dependence was clearly observed at high intensity thresholds and high temporal resolutions; the 5-min extreme intensity increases by 44% for the 99.9th and by 25% for the 99th percentile, with increasing horizontal resolution from $0.1°$ to $0.01°$. Quantitative uncertainty information from this study can guide both data users and producers to estimate uncertainty in their own observational datasets, consequently leading to the sensible use of the data in relevant applications. Our findings are generalisable to moderately hilly regions at mid-latitudes, however the degree of uncertainty could be affected by regional variations, like rainfall type or topography.

# 1 Introduction

Rainfall data are one of the most important inputs for hydrological as well as climatological studies and applications. Furthermore, fit-for-purpose information derived from rainfall data is crucial for a wider range of users, such as civil engineers, water resource managers and governments. To meet the needs of diverse user groups, rainfall observational datasets from in-situ measurement and remote sensing have been greatly enhanced in terms of both data quality and resolution (e.g., Berezowski et al., 2016; Hou et al., 2014; Keller et al., 2015; Yatagai et al., 2012). Often, rainfall data are required as areal estimates at the scale of interest, for instance, at grid or catchment scales. Point measurements from in-situ gauge observations are spatially aggregated or interpolated to estimate the areal distribution of rainfall, and hence the accuracy of areal rainfall data is highly dependent on spatiotemporal variability of rainfall events and density of observation points (Girons Lopez et al., 2015; Hofstra et al., 2010; Villarini et al., 2008; Wood et al., 2000). This limits the understanding of fine-scale rainfall processes, particularly of extreme events (Sillmann et al., 2017). Gridded rainfall data are also available from remotely sensed observations at high spatial resolutions (e.g., 1-5 $\mathrm{km}^2$ for radar data or $0.1° \times 0.1°$ for satellite data). While those data sets are good alternatives to address a number of the issues relating to the scarcity of gauges, rainfall variability at subpixel scales can still not be fully resolved (Peleg et al., 2013; Tokay et al., 2014). Moreover, systematic errors can be large, and the quality of remotely sensed data therefore strongly relies on gauge-based data that are used for their regional validation and correction (Kann et al., 2015; O et al., 2018b; Steiner et al., 1999).

Addressing the issue of spatial variability and uncertainty of rainfall has been tackled over many years with various purposes. For instance, evaluation of satellite or radar rainfall products involves investigation of larger-scale rainfall processes to assess the ability of remote sensing in capturing the inter-pixel rainfall variability (e.g., Chaudhary et al., 2017; Dhib et al., 2017; Lockhoff et al., 2014). On the other hand, small-scale rainfall processes are of interest to identify the effect of intra-pixel variability of rainfall on the performance of remote sensing (e.g., Ciach and Krajewski, 1999, 2006; Gebremichael and Krajewski, 2004; Habib and Krajewski, 2002; Peleg et al., 2013; Tan et al., 2018; Tokay et al., 2014). To quantify the rainfall uncertainty, observational data from highly dense rain-gauge networks have been employed as a ground truth. Peleg et al. (2013) used multiple rain gauges within a radar subpixel area (4 $\mathrm{km}^2$) and examined the contribution of gauge sampling error to the total radar-rainfall estimation error. Using relatively long-term gauge data (5-years), Tokay et al. (2014) analyzed the spatial correlation of rainfall for different seasons and weather systems within the footprint size of microwave satellite sensors.

A similar approach employing dense gauge networks can be adopted to diagnose the spatial variability and uncertainty of rainfall at catchment scales (e.g., 100 - 500 $\mathrm{km}^2$). Such scales are of great interest not only for the evaluation of remotely sensed data, but also for hydrological applications like runoff modelling or gauge network design. Wood et al. (2000) examined the accuracy of areal estimates of rainfall over a 135 $\mathrm{km}^2$ basin according to the HYdrological Radar EXperiment network consisting of 49 rain gauges. The network later provided a six-year rainfall dataset (from 50 gauges) for the study of Villarini et al. (2008), where a comprehensive analysis of temporal and spatial sampling uncertainties was conducted. However, most of the local areas do not have adequately dense gauge networks, which limits the comparison and verification of findings from the aforementioned studies across diverse rainfall regimes. Schroeer et al. (2018) recently employed the WegenerNet Feldbach

region (WEGN) and the surrounding operational rain gauge stations to sample summertime convective extreme events at sub-hourly to hourly scales and found a power law decay of the event maximum area rainfall with increasing interstation distance (1 km to 35 km).

In this paper, in order to contribute to the effort for better and broader assessment of the rainfall spatial variability and associated uncertainty, we employed 10-year rainfall data from the WEGN, a high-density network in southeastern Austria (Kirchengast et al., 2014). The network includes 150 rain gauges deployed over an area of $\simeq 300$ km$^2$, approximately corresponding to one gauge per 2 km$^2$. First, following previous studies (e.g., Villarini et al., 2008; Peleg et al., 2013; Tokay et al., 2014), we quantified the spatial variability of rainfall utilizing a corrologam between the gauges to understand the spatial characteristics of rainfall in the region.

Second, we investigated the uncertainty in estimating areal rainfall caused by a limited number of point observations. Given that the properties of individual rainfall events can be different from all-event averages (Ciach and Krajewski, 2006; Eggert et al., 2015), we focused on events with a potentially high impact, which we defined as the top 10% wettest days during the warm season (May–September). The accuracy of areal rainfall estimation is a long-standing issue, e.g., in catchment modelling, because error and uncertainty in rainfall data can propagate into large variations in simulated runoff, and thus it has been dealt with in diverse manners. For instance, the influence of spatial representations of rainfall input to runoff errors has been demonstrated through modelling studies (e.g., Bárdossy and Das, 2008; Xu et al., 2013). The error in catchment-scale areal mean rainfall has also been directly quantified by employing high-resolution gauge data (e.g., Villarini et al., 2008; Wood et al., 2000; Ly et al., 2011). We followed the latter approach using the WEGN rainfall data.

Finally, we compared extreme rainfall at different spatial and temporal scales using gridded rainfall fields to quantitatively assess the impact of spatial averaging on the definition of extremes. The identification of rainfall extremes based on intensity thresholds is common practice, however, the considered spatial scale of rainfall data defines different sets of extreme events (Eggert et al., 2015), potentially affecting threshold-based early warning systems (Marra et al., 2017). Although gridded datasets have been used in a range of applications like assessments of climate change impacts or evaluation of climate models, a common caveat of using the datasets in the study of extreme rainfall is that the quality of gridded rainfall data is highly constrained by the location and density of input weather station data (Hofstra et al., 2010; Prein and Gobiet, 2017). By contrast, the quasi-regular configuration of WEGN on an approximately 1.4 km x 1.4 km grid permits robust examination of the frequency and intensity of rainfall extremes at various horizontal resolutions.

Consequently, this study aims to assess spatial uncertainty of rainfall at catchment scale using rain gauge data, with a focus on heavy and extreme rainfall events. This paper is structured as follows. Section 2 describes WEGN rain gauge data and regional rainfall climatology. Section 3, Sect. 4, and Sect. 5 present the results of the data analysis. We close with discussion and conclusions in Sect. 6.

## 2 WEGN rainfall data and regional rainfall climatology

The 10-year rainfall data (2007-2016) are obtained from the WEGN Feldbach region network in southeastern Austria (Kirchengast et al., 2014). Of 154 weather stations, 150 stations that are equipped with tipping-bucket rain gauges are used in this study (Fig. 1). Raw rain gauge data are aggregated every five minutes. Errors in the rainfall data were comprehensibly analysed and corrected by O et al. (2018a). The gauges are almost uniformly spaced over an area of 20 km × 15 km with moderate topography (about 260 to 520 m altitude). The inter-gauge distances range from approximately 0.7 km to 23.4 km. The gridded fields of rainfall are constructed by an inverse distance weighting (IDW) on a 200 m × 200 m Universal Transverse Mercator grid. WEGN station and gridded data products are available at www.wegenernet.org.

Southeastern Austria including the Feldbach region is influenced by both continental and Mediterranean climates. The region receives high amounts of rainfall during summer months. The occurrence of thunderstorms and hail is higher than in other parts of Austria (Matulla et al., 2003). Figure 2 shows average diurnal variations of rainfall and temperature over the entire network during the study period. The WEGN area is characterized by warm and wet months from May through September (hereafter "warm season") and relatively cold months without much rainfall during the remaining seven months (hereafter "cold season"). The average monthly rainfall is 102.8 mm in the warm season, while 48.9 mm in the cold season. The diurnal signal is more clearly seen in the warm season for both rainfall and temperature. Rainfall maxima occur often in the early afternoon through midnight, shortly after maximum temperature, implying that a major contribution to the warm season rainfall is from short-duration convective events. Because diurnal heating plays an important role in triggering thermal convection, most inland regions show afternoon rainfall maxima (Dai et al., 2007). Extreme daily precipitation, however, can also be caused by Genoa Lows, which transport humidity from the Mediterranean Sea, yielding intense rainfall with long-duration.

## 3 Spatial variability of rainfall

The spatial structure of rainfall events is studied using the Pearson's correlation coefficient between all pairs of rain gauges. Pearson's $r$ is the most commonly used rainfall correlation estimator (e.g., Ciach and Krajewski, 2006; Jaffrain and Berne, 2012; Peleg et al., 2013; Tokay et al., 2014; Villarini et al., 2008). At sub-daily and daily timescales from 5-min to 24-h (06-06 UTC), the correlation of rainfall among rain gauges is calculated for each year. One year period includes a set of warm season (May to September) and cold season (October to next April). The incomplete years (i.e., first and last years) are excluded from the calculation of all-months (May to next April), whereas the warm and cold seasons have 10 annual curves each. The data pairs when both record zero rainfall are discarded. The correlation values in each period were then sorted according to the separation distance of gauge pairs and averaged into the nearest 1-km distance bins. We fitted a three-parameter exponential function to the average correlations. The distance bins for fitting the model were taken up to and including 15 km given the network dimension, which means that rainfall data pairs were sampled uniformly for any spatial direction. The spatial correlation ($r$) at separation distance $h$ is:

$$r(h) = c_1 \, exp\left[-\left(\frac{h}{c_2}\right)^{c3}\right] \tag{1}$$

where $c_1$ represents the nugget effect, $c_2$ is the correlation distance, and $c_3$ is the shape factor. The parameters are determined by least-squares curve fitting. Figure 3 shows the spatial correlation of all-months, warm, and cold seasons for four selected accumulation times. A logarithmic transformation is applied to the data; $log(x + 1)$ to keep zero rainfall, where x is in rainfall in mm. As the transformation make rainfall data conform more closely to the normal distribution, the effects of extreme values on correlation coefficients is mitigated (Habib et al., 2001; Jaffrain and Berne, 2012). This results in slightly lower correlations (not shown), however, the overall pattern of correlation decay curves remains unaffected. The data after the log-transformation are used in the figure.

Many factors are known to affect the spatial correlation structure in rainfall. For instance, Habib et al. (2001) examined the sensitivity of correlation estimation in rainfall to sample size or extreme rainfall events and Huff and Shipp (1969) demonstrated how the rate of correlation decay varied with different rainfall types. We therefore do not make a direct comparison of correlation values with those from other studies, yet we still observe that the behaviors of the correlation decay found in this study are in broad agreement with spatial rainfall correlation structures reported in the aforementioned studies. First, longer accumulation times show higher $c_1$ (i.e., smaller microscale variations) and longer correlation distance values. Second, short-range correlation decreases rapidly with increasing separation distance, particularly at sub-hourly scales.

The warm season shows higher spatial variability of rainfall compared to the cold season, due to a higher proportion of convective events. The correlation curves of all-months show a more similar pattern with the warm season, as expected, given that most of the rainfall events are concentrated during the warm season (see Sect. 2). Tokay et al. (2014) found substantial year-to-year variations especially during autumn and spring. Similarly, WEGN rainfall shows marked interannual variability, but also during the warm season. It should be noted that the correlation functions of the cold season start with lower $c_1$ values than of the warm season, meaning larger measurement errors and microscale variability of rainfall. This could be related to winter precipitation types in the region. For instance, uncertainty affecting the gauge measurements (e.g., wind-induced bias) may play a bigger role in determining the spatial heterogeneity of neighboring stations during low-intensity precipitation events, than during warm season convective events. Another possible reason is that WEGN does not accurately capture solid precipitation (O et al., 2018a), since only few gauges are heated, and thus systematic errors between neighboring gauges can be greater during the cold season, possibly yielding the low $c_1$ values.

Figure 4a-c summarizes the time dependence of the three parameters. Synthesized parameters here are obtained from the fitting function that is constructed by averaging yearly correlation values in each distance bin. Nugget effect values range from 0.71 to 0.98 for the cold season, while from 0.85 to 0.99 for the warm season. The correlation distance of the cold season at the 3-h is nearly corresponding to the correlation distance at the 24-h scale in the warm season. The parameter values of all-months are located between those of warm season and cold season. We found that the dependency of nugget effect and correlation distance on times scale is similar to the results by Villarini et al. (2008). The nugget effect parameter changes sharply at smaller timescales, while the correlation distance appears to be more sensitive for larger timescales. The shape factor of this study, however, does not show a clear increasing or decreasing trend. This is consistent with findings from Peleg et al. (2013) and Tokay et al. (2014). We selected the three-parameter model for the function fitting, because the model shows the minimum root-mean-square error (RMSE) between observed and fitted correlation values across all time scales among the several tested

models (Fig. 4d). Note that we fitted the correlation models to bin-averaged values and thus obtained relatively small fitting errors compared to other studies (e.g., Ciach and Krajewski (2006) or Tokay et al. (2014)). During multiple tests with different fitting models, we found that the fitted correlation distances over 100 km (e.g., values at accumulation times of >6-h in Fig. 4b) are often highly impacted by the selected fitting models. However, this model uncertainty does not affect the general behaviors of the parameters including their dependence on time scale and their seasonal differences. Nonetheless, when the spatial scale of observed correlations is limited to a distance of a few kilometers (e.g., 15 km in our study), the correlation distances estimated from the fitting model should be interpreted with caution. Interested readers may obtain a more detailed discussion of the fitting model in Svoboda et al. (2015).

So far we assume that the correlation structure is isotropic. To check the directionality (anisotropy) of the spatial correlations, we remapped them onto the two-dimensional space (Velasco-Forero et al., 2009; Mandapaka and Qin, 2013). Fig. 5 shows the 10-year averaged correlations plotted over 1 km $\times$ 1 km grid cells but only till the e-folding distance (when the correlation drops to around 0.37). While the correlation drops rapidly in all directions over short distances, a strong correlation is observed in an approximate southwest–northeast direction as separation distance increases. The directionality is more pronounced during the cold season, which can be interpreted as a consequence of movement of large-scale weather systems (contrary to summertime convective) along the favoured wind direction during the season, rather than as an effect of orographic barriers. Such directional characteristics of the correlations are averaged out in Fig. 4.

# 4 Accuracy of areal rainfall estimation during heavy rainfall events

In this section we investigate data uncertainty associated with areal rainfall estimation. In particular, the study focuses on high-impact rainfall events. While heavy rainfall is one of the major hydrological hazards, its accurate spatial representation over an area remains a subject worthy of inquiry. Heavy rainfall events are defined as days with total rainfall exceeding the 90th percentile of the daily rainfall, without a consideration of rainfall type. Only the warm season is considered. As a result, a total of 71 days are selected. According to our visual-inspection on rainfall time series, the selected days likely include mixed rainfall types (short- and long-duration rainfall) rather than a specific type. The median of gauge-averaged accumulations is 28.1 mm d$^{-1}$, with a range of 19.8 mm d$^{-1}$ to 64.1 mm d$^{-1}$ (more information is given in Appendix A).

We assume that the mean areal rainfall of a full density network represents the "truth". The areal rainfall of $n$-gauge networks ($n$ = number of gauges) is calculated and compared with the true rainfall to quantify the accuracy of areal rainfall estimation with low-density networks (see also Villarini et al., 2008). Each $n$-gauge network consists of randomly selected 1,000 possible gauge combinations. The 1-gauge network has 150 cases. As shown in Fig. 6a, the average and spread of normalized RMSEs (NRMSEs) of areal rainfall estimation tend to decrease with rising gauge number. The mean number of gauges required to obtain areal rainfall with NRMSEs lower than 20% is given as a function of time resolution in Fig. 6b. The curve (in black) roughly exhibits power-law behavior; $74.2 \times t^{-0.4}$, where $t$ is the time resolution (minute). At the daily scale, more than one gauge per 300 km$^2$ would be sufficient to reach a <20% estimation error. Correspondingly, at the temporal scales of 1-h, 30-

min, and 5-min, on average more than 12, 18, and 33 gauges, respectively, are needed to achieve the same level of accuracy. Villarini et al. (2008) found that four gauges are necessary at the daily scale for the same accuracy level for an area of 135 km$^2$. Heavy events are not explicitly considered in their study.

One should note that the use of randomly selected gauge combinations only offers a rule of thumb about the required number of gauges to minimize uncertainty in areal rainfall estimates. To demonstrate the role of gauge distribution in determining the estimation error, we selected 'good' and 'bad' distributions, 100 cases, respectively, out of the 1,000 combinations for each $n$-gauge networks that ranked in the top 10% and bottom 10% based on the area-of-influence (see Appendix B). As seen in Fig 6a (red crosses), the smallest estimation error is obtained with regularly distributed gauges. In other words, a well-designed gauge network allows to meet the desired error limit with a smaller number of gauges (grey curve in Fig 6c). For example, at a 1-h scale, the 20% estimation error can be reached using uniformly distributed 8 gauges, however, the same level of accuracy cannot be guaranteed even with 23 rain gauges if their spatial configuration is not properly structured.

We repeated the calculation of the required gauge number to reach the certain accuracy, using sub-areas of 150, 100, and 50 km$^2$, i.e., 1/2, 1/3, and 1/6 of WEGN area size, respectively (gray lines in Fig 6b and c). For each sub-area, the mean rainfall of all gauges within the area is assumed to be truth and the 1,000 possible gauge combinations are randomly selected, as we did above. For the 50 km$^2$ area where only 25 gauges are available, all combinations are included when the total possible combinations of $n$-gauges are less than 1,000 cases. For any case, the dependence of the accuracy of areal rainfall estimates on the gauge number shows the power-law behavior across time scales. However, the required gauge numbers do not linearly decrease as the considered network area decreases. For smaller areas, we need more number of gauges per km$^2$ (i.e., higher gauge density) to reach the same level of accuracy at the same time scale (see inset plots in Fig 6b and c). Because rainfall variability varies faster within the first few kilometer, more dynamic rainfall variations in the smaller areas cannot be properly captured when the inter-distances of gauges remain the same (i.e., constant gauge density between the sub-areas), particularly for short time scales.

Additionally, the effect of gauge density on event-based rainfall statistics is assessed in Fig. 7. Daily rainfall accumulation and peak hourly rainfall of the 71 heavy daily events are recalculated using predefined sub-networks with gauges ranging from 1 to 16. The gauges are uniformly spread; the definition of the sub-networks can be found in Appendix B. While the sub-network with only one gauge exhibits large overestimation errors for both total and peak rainfall, employing an additional gauge already significantly reduces the degree of errors and yields underestimation error more frequently than overestimation. Note that Austrian weather service (ZAMG) has two operational stations over the actual WEGN area. Given that convective storms occur on scales of a few kilometers, low-density gauges over the region are likely to miss the core of storm. On the contrary, low-density gauges can also overestimate rainfall intensities by capturing only the core of storm, but the magnitude and frequency of these errors appear slightly less than those of the underestimation errors. There is no significant difference in either average error or spread of errors from more than 10 gauges, as expected from Fig. 6.

## 5    Impact of spatial aggregation on extreme rainfall

We next focus on the uncertainty of area- or grid-averaged rainfall relating to spatial data resolution for the heavy rainfall events. Figure 8 compares rainfall percentiles among the gauges. Grey lines mean a 10-90th percentile range of rainfall intensities at a given percentile bin. For example, at the 30-min scale, the 99.9th percentile (the top 0.1%) rainfall intensity corresponds to roughly 45 $\mathrm{mm\,h^{-1}}$ at most gauges, while it exceeds 52 $\mathrm{mm\,h^{-1}}$ at certain gauges. It is also seen that 10% of WEGN gauges (i.e., 15 gauges) records are found to be lower than 38 $\mathrm{mm\,h^{-1}}$. The upper tail of the rainfall distribution shows strong spatial variation. Such point-scale extreme rainfall features will be completely missed unless there exist dense rainfall observations, or they are inherently smoothed out in gridded data.

In fact, many studies have pointed out that the use of gridded rainfall data can lead to erroneous analyses of small-scale extremes because of the limited number of point observations (Contractor et al., 2015; Hofstra et al., 2010; Prein and Gobiet, 2017; Tozer et al., 2012). In addition to the high-resolution, the regular distribution of WEGN gauges enables generating gridded rainfall fields that are homogeneous in space, and, consequently, robustly assessing uncertainty in rare and extreme rainfall represented in the data.

We generated gridded data using all 150 WEGN gauges and rescaled the data into horizontal resolutions from 0.01 to 0.1 degree (hereafter HR01 to HR10). Spatial aggregation begins from the top-left corner towards the bottom right and the remaining southern and/or eastern part of the grid is discarded (see Fig 10). HR01 corresponds to about 1.1 km and 0.8 km in latitudinal and longitudinal directions, respectively. Figure 9 shows the 99.9th and 99th percentiles of heavy rainfall intensities as a function of space-time scales. Although temporal aggregation more significantly alters the definition of extremes, the impact of spatial aggregation is also notable, particularly at the sub-hourly scales. The 5-min extreme intensity decreases from HR01 to HR10 by 30% for the 99.9th percentile while it decreases by 20% for the 99th percentile.

Meanwhile, although the spatial aggregation impact is much less pronounced at a daily scale, the selected spatial scale still affects statistics of extreme areal rainfall, such as daily extreme frequency. This is shown in Fig. 10, which illustrates the occurrence of days above a selected threshold; top 5% of heavy rainfall events at HR01. The concept of the exceedance probability above thresholds is widely used in analyses of rainfall-triggered risk. Some HR01-scale sites appear to experience extreme rainfall more frequently than other part of the region. In other words, high-resolution data well-represent spatial variation and frequency of rainfall extremes, neither of which is seen in lower-resolution data. Many existing gridded datasets are not likely to fully sample such site-level extreme events owing to limited spatial resolution. The exceedance probability of extreme rainfall across spatial resolutions is given in Fig. 11. The impact of different data resolutions on extreme rainfall occurrence is pronounced in both lower and upper tails. The highest daily rainfall during 10-years appears to be 68.4 $\mathrm{mm\,day^{-1}}$ at HR10, but 104.4 $\mathrm{mm\,day^{-1}}$ at HR01; the maximum record over the entire WEGN area is 64.1 $\mathrm{mm\,day^{-1}}$, so the ratio of the site-to-areal extreme rainfall ranges from 1.07 to 1.63 depending on the considered spatial scale.

## 6  Discussion and conclusions

The understanding of spatial uncertainty in heavy rainfall at fine scales has been hampered by the limited availability of suitable and reliable observational datasets. Although high-resolution radar data are often used to study small-scale rainfall variability, the use of radar data is dubious, as indicated by Svensson and Jones (2010), owing to their indirect measurements of rain

and relatively short records. In this study, we used 10-year rainfall measurement data from the 150 rain gauges, uniformly spaced over the WEGN network in southeastern Austria. First, to quantify rainfall variability, spatial correlation between the gauge records was examined. We found that the degree of spatial rainfall variability can be substantially different not only within years (warm versus cold seasons) but also between years. This implies that long-term data should be considered to obtain comprehensive perspectives on regional rainfall variability. In fact, individual weather systems can exhibit varied spatial

characteristics (Habib and Krajewski, 2002; Ciach and Krajewski, 2006; Tokay et al., 2014). In southeastern Austria, including the WEGN area, Schroeer et al. (2018) found much steeper decay in a correlogram function when only extreme summertime convective events are accounted for. We also found that during the cold season, the density of gauges is less of a concern (showing longer correlation distance) compared to the warm season. However, low values of the nugget effect parameter imply that snow measurements during winter time remain a challenge, especially at short time scales. Additionally, we checked the

directional features of the spatial correlations. The spatial correlations in the southwest–northeast direction decay faster than in the southeast–northwest direction. This anisotropic pattern appears to be more pronounced over large separation distances during the cold season, which is probably linked to large-scale weather systems. This indicates that the assumption of isotropic correlations can be another source of uncertainty, under certain weather conditions, when modeling spatial correlation of rainfall (especially for estimation of correlation distance).

Secondly, we demonstrate how high density and regular distribution of WEGN gauges contribute to delivering accurate areal-precipitation estimation. The overall uncertainty in mean areal rainfall shows a clear dependence on the number of gauges and the temporal resolution considered for the estimation. To reach the same level of accuracy, the average number of gauge has to be increased roughly following a power law as time scale decreases. Given that only two operational meteorological stations exist over the WEGN area, the insufficient gauge density may hamper the use of the station data to construct spatial rainfall

fields in the region, especially at sub-daily scales. Further analysis shows that there is no linear relation between the required number of gauges and the ratio of considered area size. The accuracy of areal rainfall estimation is also significantly dependent on the spatial configuration of the network. Assuming that we have a well-distributed gauge network, it is observed that at least 2-5 gauges are required in the WEGN area ($300 \ \mathrm{km}^2$) for accurate areal rainfall estimates such that we can obtain reliable rainfall event statistics (e.g., total amount and peak hourly intensity of daily heavy rainfall events) with no significant error.

When more than 10 gauges are available in the area, the impact of gauge number or configuration on the spread and mean of errors in areal rainfall estimation becomes marginal. Our findings have implications concerning the use of sparse observational gauge data, for instance, in hydrologic modeling or rainfall estimates evaluation (e.g., Syed et al., 2003; Tian et al., 2018).

Lastly, using gridded WEGN data, rainfall extremes are reproduced at multiple spatial scales; approximately from the grid resolution of regional to convective-permitting models (about $11.1 \ \mathrm{km}$ to $1.1 \ \mathrm{km}$ in latitudinal direction). We show how

different rainfall events can be considered extreme depending on the spatial and temporal resolutions. We found that the 5-min extreme intensity increases by 44% for the 99.9th and by 25% for the 99th percentile, when the horizontal resolution increases from 0.1° to 0.01°. The results also demonstrate that high-resolution gridded data provide more reliable information not only in terms of the magnitude and frequency of extremes, but also in terms of the exact location of the extremes. As a result, limited resolution of rainfall data can alter interpretations of rainfall statistics; extreme rainfall events at a location of interest (a $0.01° \times 0.01°$ site in our example) could occur more frequently and more intensely versus the local average. For instance, the highest daily rainfall during 10 years appears to be $68.4 \, \mathrm{mm \, day^{-1}}$ at 0.1°, but $104.4 \, \mathrm{mm \, day^{-1}}$ at 0.01° resolution. Localized information from high-resolution observation is the key for developing prevention and protection plans to mitigate potential damages of extreme rainfall in an efficient and adequate way. Our results highlight the need to evaluate uncertainty in extreme statistics derived from the existing datasets for supporting data selection among available rainfall data products.

In conclusion, the WEGN network provides a unique opportunity to empirically assess spatial variability and uncertainty of surface rainfall directly based on gauge data. The network provides long-term records, of more than a decade, which permits to exclusively focus on heavy rain events. Nonetheless, as stated in Villarini et al. (2008), there are only a few dense gauge networks on the catchment scale, so the verification of findings from studies in other regions is challenging. Regional variations, such as topography or rain type, can lead to differences in the degree of rainfall variability and uncertainty (e.g., Buytaert et al., 2006; Prein and Gobiet, 2017). Therefore, some of the general conclusions of this study may only be representative for mid-latitude regions with moderate topography. In addition, more robust interpretation of the rainfall spatial structure beyond the network dimension (> 15 km) needs to be complemented by additional larger-scale gauge data. For instance, Schroeer et al. (2018) used three different scales of networks, including the WEGN, to estimate the underestimation of maximum area precipitation of extreme convective over the range of 1 km to 30 km. It should be noted that WEGN has a high flexibility in terms of providing rainfall data within various spatial scales thanks to both high-resolution and quasi-grid configuration of the gauges. In this context, WEGN will continue providing observational evidence to explore local-to-catchment scale rainfall processes over the next years.

*Data availability.* WegenerNet data products are available at www.wegenernet.org.

## Appendix A: Heavy rainfall events

Table A1 gives general information about the selected heavy rainfall events studied in Sect. 4 and Sect. 5. The events are corresponding to the days when total rainfall is greater than the 90th percentiles of daily rainfall (06-06 UTC), during the warm season. Peak ratio is given as a ratio of peak hourly rainfall to daily total. Rainfall in the region during the summer months is triggered by the advection of humid air masses from the Adriatic Sea. Heavy rainfall events are closely linked with local thunderstorms (Matulla et al., 2003, see also Sect. 2). The rain type is not explicitly considered for the event selection.

## Appendix B: Definition of rain-gauge sub-networks

Figure A1 shows the selection order of WEGN gauges for defining the low-density sub-networks that were used in Fig. 7 of Sect. 4. Priority consideration was given to the actual location of operational weather stations within the WEGN network; the selected gauges 1 and 2 are located nearest to the member stations of the Austrian weather service (ZAMG) and the gauges 3, 4, and 5 are nearest to the rain gauges operated by the Austrian hydrographic services (AHYD). The gauges afterward were arbitrarily selected, ensuring a spatially uniform distribution. Normalized standard deviation of area-of-influence was used as an index for the uniformity of gauge configuration, which fluctuated between 0.37 and 0.23 with a decreasing trend as the number of the selected gauges increases. The area-of-influence is defined as follows: small grid boxes (approx. $0.01° \times 0.01°$, a total of 406 boxes) were defined over the WEGN network and each box is assigned to the nearest gauges of a given sub-network. Then, with an assumption that the most regular gauge configuration would share the same number of boxes, standard deviation of the area-of-influence of $n$-gauges is calculated. For instance, for the $five$-gauges sub-network, each gauge is expected to share around 80 boxes under an ideal situation. However, in this study, the five gauges share 71 to 113 boxes each, resulting in the uniformity index of 0.35. Note that this simple method does not consider the degree of centralization. The uniformity index defined here is also used for Fig. 6 to select well- and badly-distributed $n$-gauge networks.

*Competing interests.* The authors declare that they have no conflict of interest.

*Acknowledgements.* The authors thank G. Kirchengast and J. Fuchsberger (University of Graz) for fruitful discussions. The study was funded by the Austrian Science Fund (FWF) under research grant W 1256-G15 (Doctoral Programme Climate Change Uncertainties, Thresholds and Coping Strategies). WegenerNet funding is provided by the Austrian Ministry for Science and Research, the University of Graz, the state of Styria (which also included European Union regional development funds), and the city of Graz; detailed information is found at www.wegcenter.at/wegenernet.

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

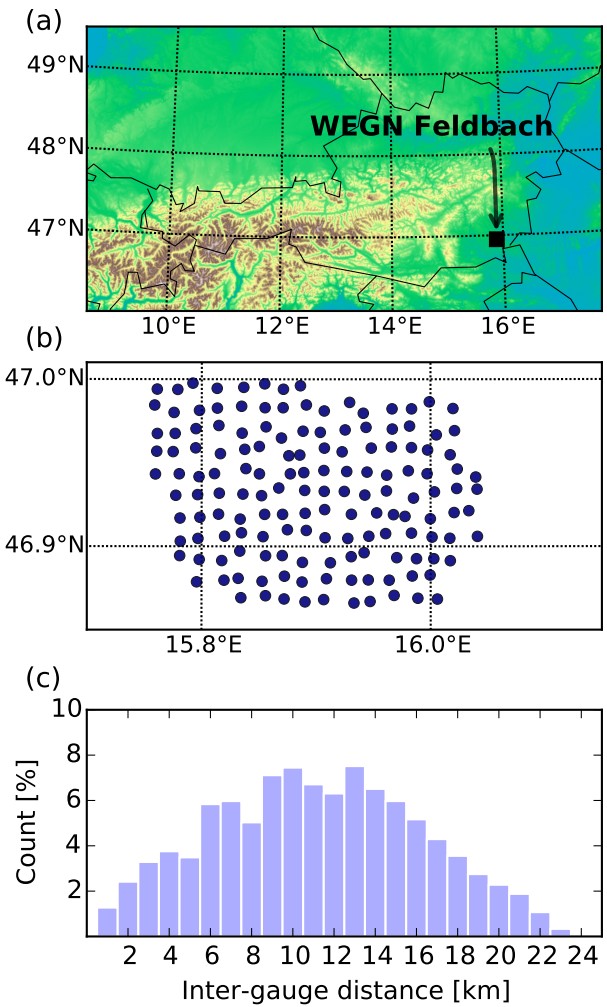

**Figure 1.** (a) WegenerNet Feldbach region (WEGN) network in southeastern Austria, (b) location of 150 tipping-bucket rain gauges, and (c) inter-gauge distances, rounded to the nearest 1-km bins.

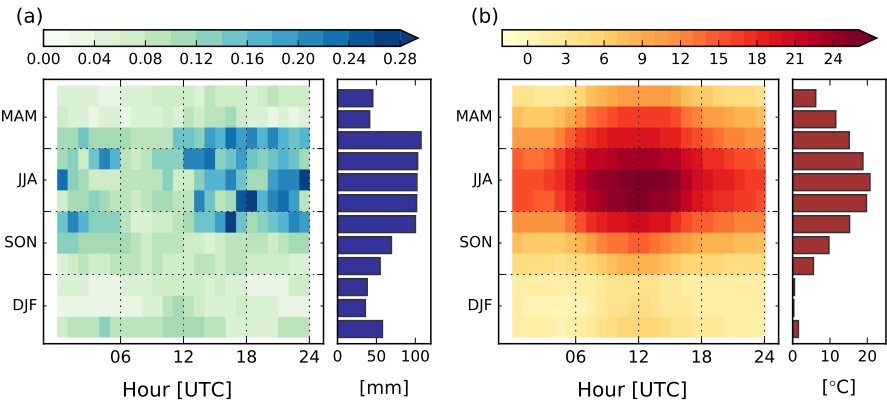

**Figure 2.** Diurnal cycles of (a) rainfall and (b) temperature derived from WEGN observational data.

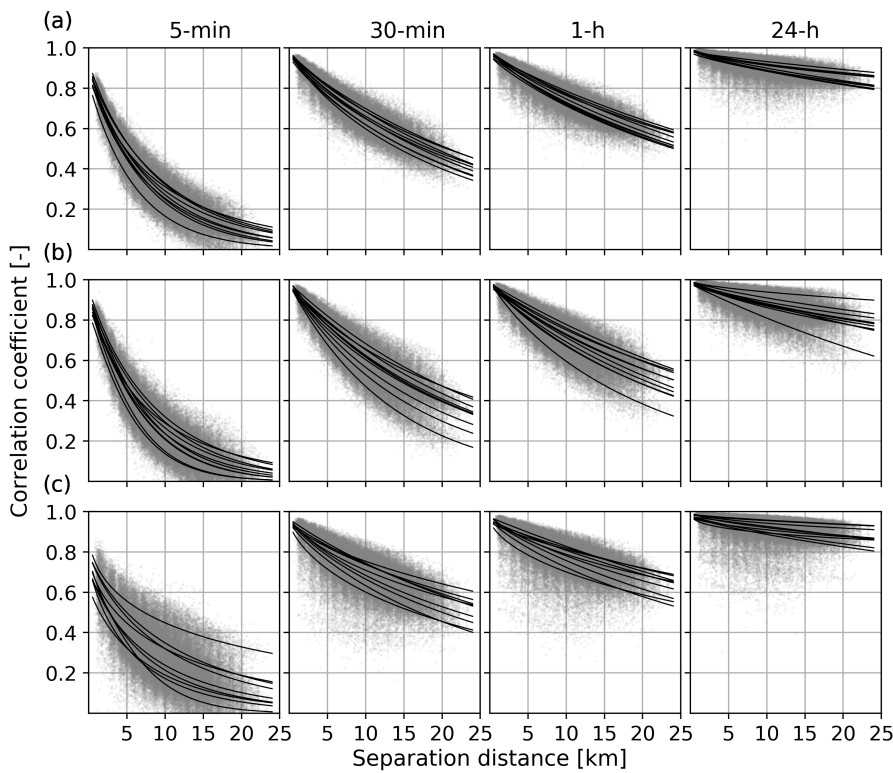

**Figure 3.** Spatial correlation of rainfall among rain gauges for (a) all-months, (b) warm season, and (c) cold season. Four selected accumulation times are shown. Each solid line represents a fitted exponential function for each year, up to 23 km (the longest inter-gauge distance of WEGN). Note that the function is fitted to correlation for separation distances <= 15 $km$ to sample data uniformly for any spatial direction.

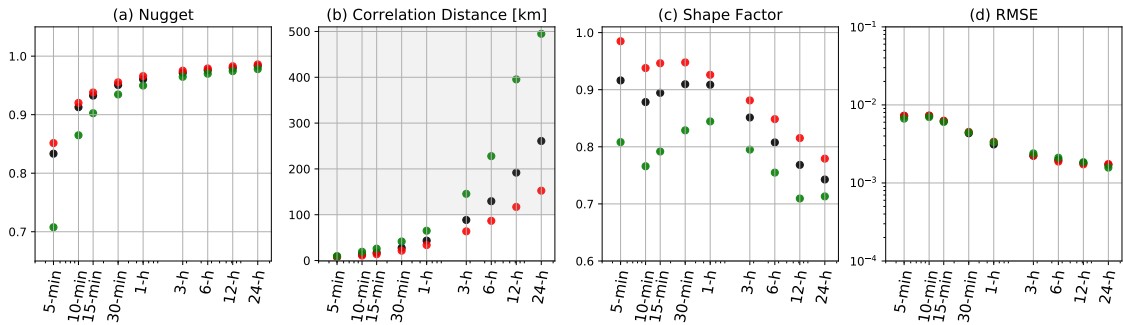

**Figure 4.** Dependence of (a) nugget effect, (b) correlation distance, and (c) shape factor of the fitted exponential functions on timescale. (d) shows RMSE of fitted correlation values compared to observed values (red: warm season, green: cold season, black: all-months). Note that the correlation distance values over 100 km (shaded in gray) are highly affected by the model selected to estimate the values.

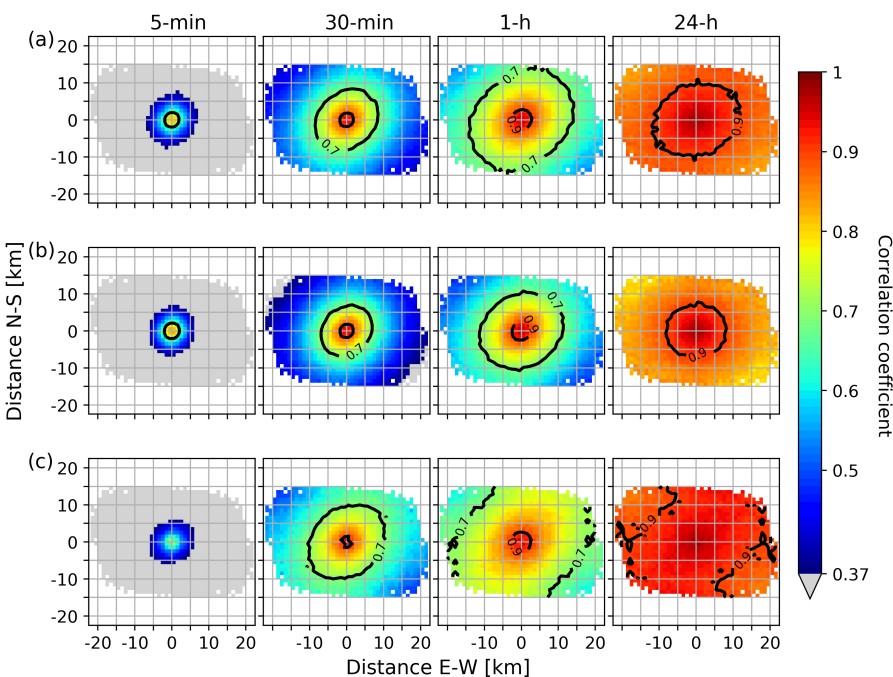

**Figure 5.** Same as Fig. 3, but for 10-year averaged spatial correlations in a two-dimensional space which is defined according to the distance between the gauges in east–west direction (x-axis) and north–south direction (y-axis). (a) all-months, (b) warm season, and (c) cold season are shown. The values beyond the e-folding distance ($\approx$0.37) are colored grey.

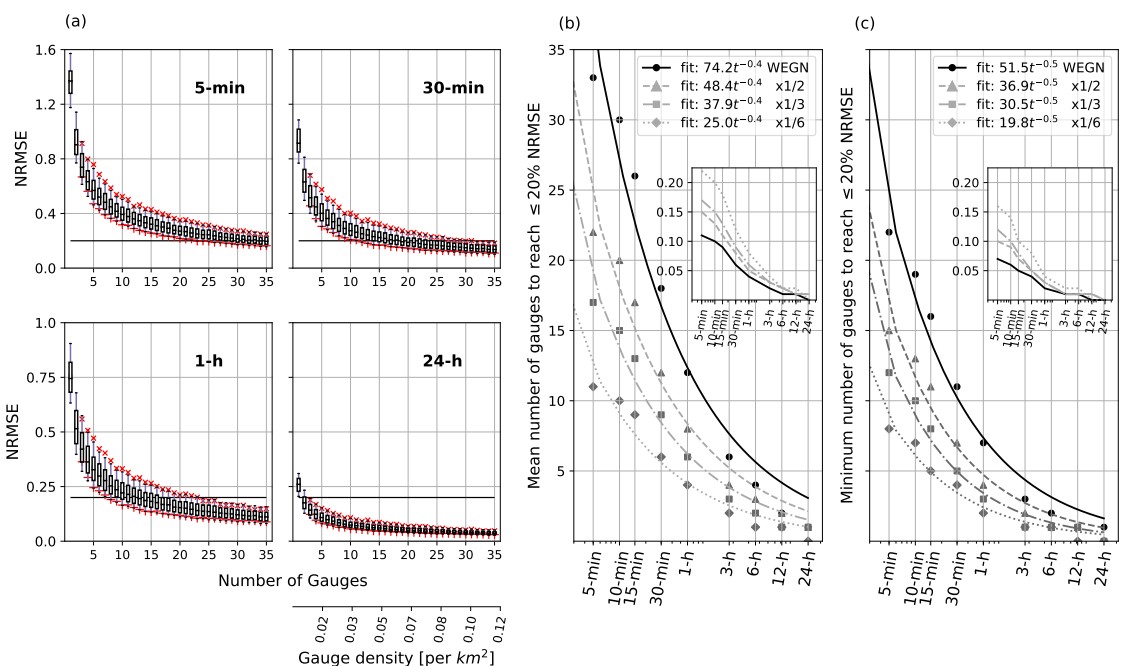

**Figure 6.** (a) Dependence of the accuracy of WEGN areal rainfall estimates on the number of gauges during heavy rainfall. Normalised RMSEs (NRMSEs) of 1,000 random gauge combinations are used to assess the accuracy for each $n$-gauge network. Four selected time accumulation are shown. Black horizontal lines correspond to 20% NRMSE. Box plots display the median, 25th and 75th percentiles of NRMSE distribution, and whiskers extend to the 10th and 90th percentiles. Red crosses and Xs show the median NRMSE for good and bad gauge configurations; 100 cases are selected, respectively, for each of the 1,000 combinations. (b) The average number of gauges required to obtain areal rainfall estimates with the NRMSE < 20% within the whole WEGN area (black) and within the WEGN×1/2, ×1/3, and ×1/6 areas (gray). Inset shows the results in term of gauge density. (c) Same as (b), but for the minimum number of gauges require to obtain areal rainfall estimates with the NRMSE <20%. Note that (a) shows the results with respect to the whole WEGN area.

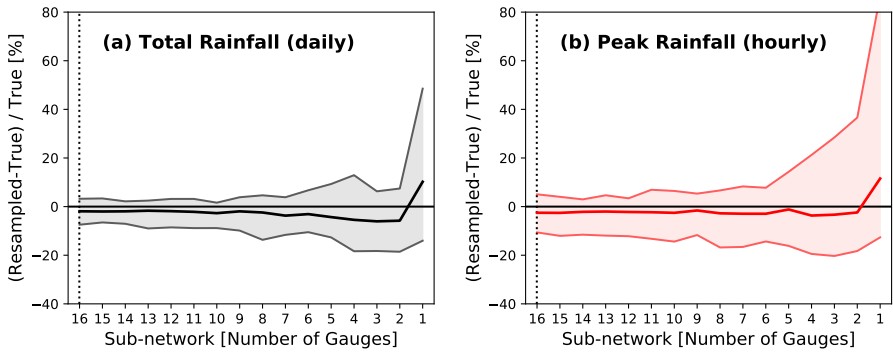

**Figure 7.** Dependence of the accuracy of (a) daily rainfall and (b) hourly peak intensity on the number of gauges. 71 heavy rain events are considered. The y axis displays the relative difference between resampled and true rainfall. Resampled rainfall is calculated from $n$-gauge sub-networks, while true rainfall is calculated using the full density WEGN network. The thick lines show the median and the shaded areas show the 10th to 90th percentile spread.

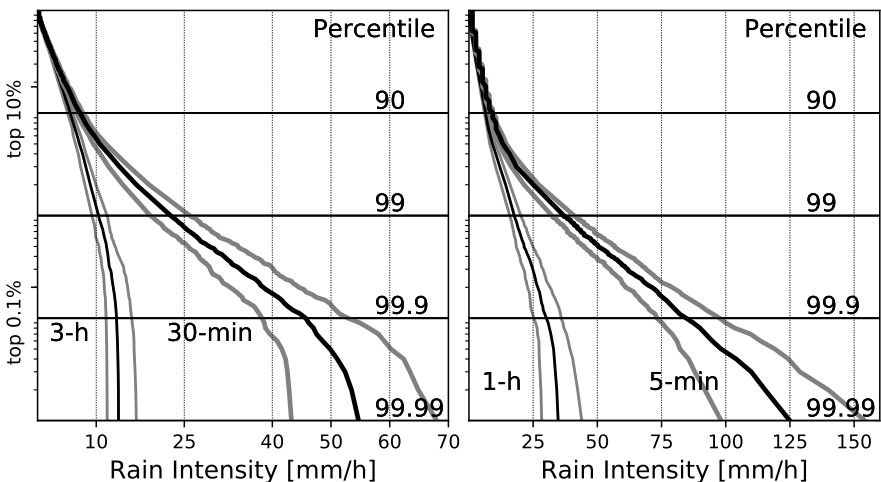

**Figure 8.** Distribution of gauge-level rainfall intensities corresponding to given percentile thresholds during heavy rainfall events. Four time scales are selected. Black lines show median values, gray lines show a 10th-90th percentile range among the gauges at a given threshold bin.

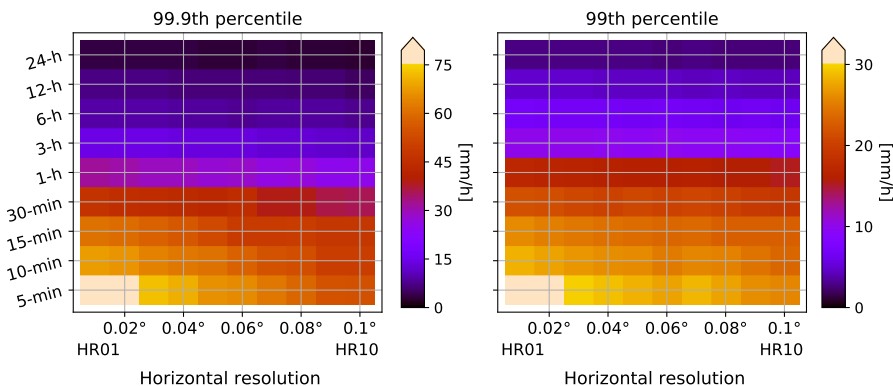

**Figure 9.** 99.9th and 99th percentiles of rainfall intensities derived from gridded rainfall fields with different spatial and temporal scales. Note that different color scale for the plots.

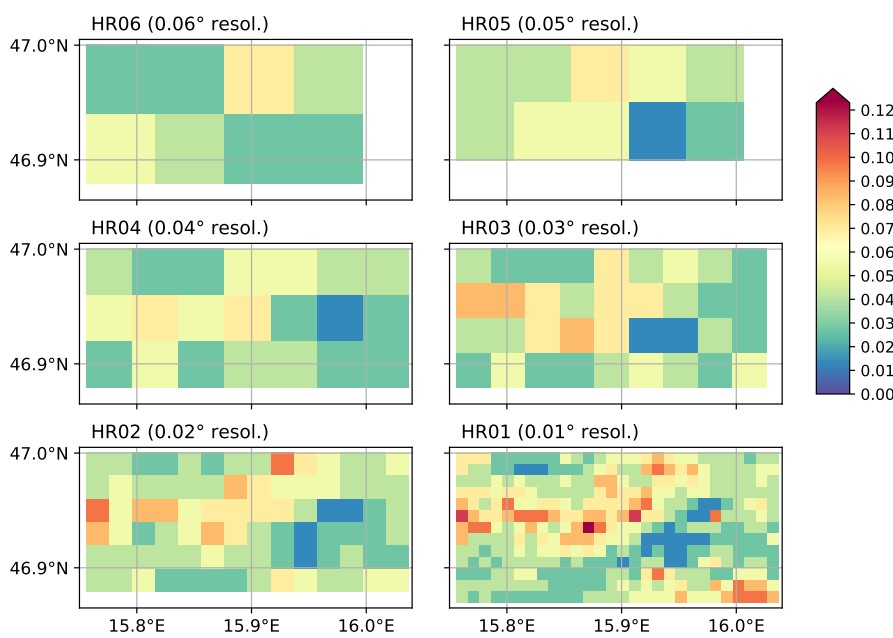

**Figure 10.** Occurrence of extreme events (defined as days with total rainfall $\geq$ 95th percentile of daily rainfall intensity during heavy rainfall events at HR01) at different horizontal grid spacing.

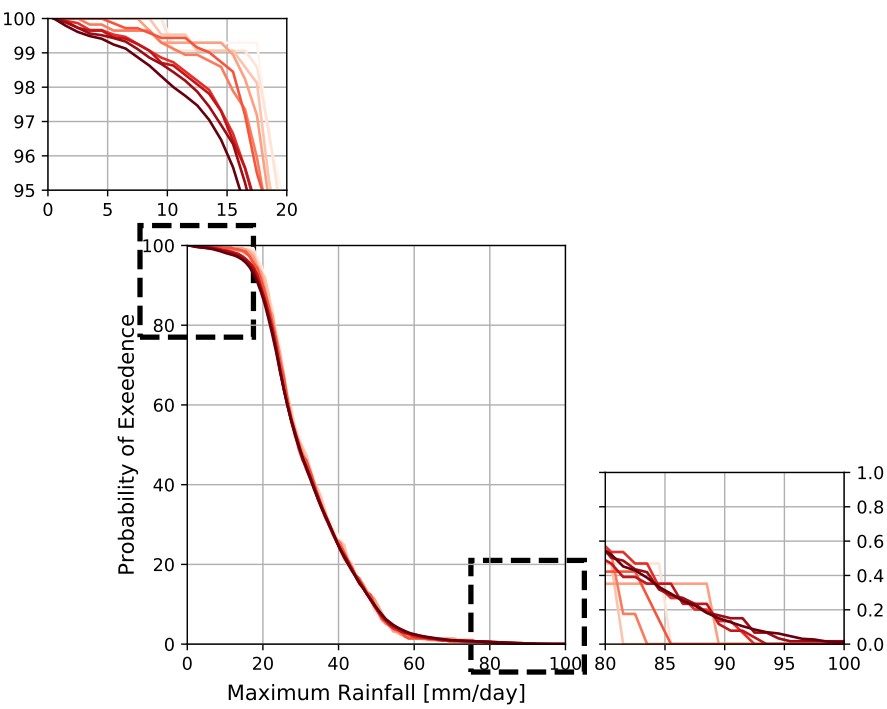

**Figure 11.** Probability of occurrence of heavy rainfall for different horizontal resolutions. Darker red represents higher horizontal resolution (from 0.1° to 0.01°).

**Table A1.** Information of selected heavy rainfall events

|  | Min | Median | Max |
|---|---|---|---|
| Total rainfall $(\mathrm{mm\,d^{-1}})$ | 19.8 | 28.1 | 64.1 |
| Peak hourly rainfall $(\mathrm{mm\,h^{-1}})$ | 2.6 | 8.6 | 26.2 |
| Peak ratio (Peak hourly rainfall/Total rainfall) | 7.8 | 25.4 | 91.0 |
| Duration $(\mathrm{h})$ | 2.0 | 9.5 | 22.5 |

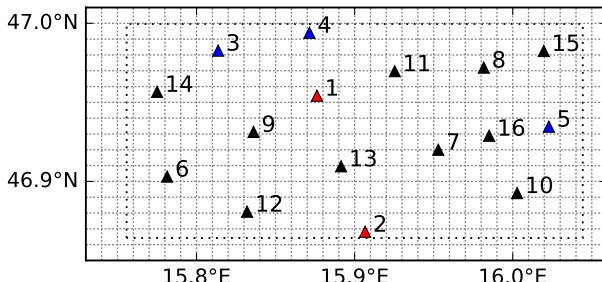

**Figure A1.** Selected WEGN gauges for Fig. 7. The gauges nearest to operational weather stations of the ZAMG and AHYD are in red and blue, respectively.