# Peer review of "Assessment of spatial uncertainty of heavy rainfall at catchment scale using a dense gauge network"

_Hydrology and Earth System Sciences, 2018_

## Referee Comment (RC1) · Anonymous Referee #1 · 5 Nov 2018

Review of the manuscript #2018-517: "Assessment of spatial uncertainty of heavy local rainfall using a dense gauge network"

Authors: Sungmin O and Ulrich Foelsche

General remarks:

The study analysed uncertainties in the estimation of areal rainfall from point rainfall measurements using data from a dense gauge network in Austria. The study is interesting especially because it presents results from a dataset with unique characteristics such as high density (150 gauges in 300 sq. km), long record (10 years), and pseudo-uniform configuration (gauges located on an approx. regular grid). The topic addressed

in the manuscript is within the scope of HESS. However, the manuscript would require considerable revision, before it can be accepted for publication. Following are my comments, which are mainly regarding the interpretation and presentation of results, and as such, can be addressed without much recomputations.

Major comments:

1) Discussion on the required number of gauges: The study provides information on the number of gauges required to estimate areal rainfall within an estimation error (NRMSE) of 20%. Such information would be useful for many purposes including i) the design of new gauge networks, and ii) quantifying estimation errors from the existing networks. However it should be noted that it is BOTH the number and the spatial configuration of the network that determine the estimation error. The interplay between spatial configuration and number of gauges can be easily illustrated using hourly scale results in Figure 5 (bottom-left panel). The authors conclude here that 12 gauges are required at hourly scale for NRMSE to be < 20%. However, from Figure 5, it can be seen that the NRMSE at hourly scale can be brought to within 20% even with 8 gauges, although for a specific spatial configuration. So, from the perspective of gauge network design, meeting the desired error limit with the minimum number of gauges is more useful. On the other hand, with 24 gauges, the NRMSE at hourly scale will be below 20% for all 1000 spatial configurations considered in Figure 5. In short, 20% NRMSE at hourly scale can be achieved with 8-24 gauges depending on their spatial configuration. On average (i.e. not always), 12 gauges ensure <20% NRMSE. I suggest that the authors highlight the role of spatial configuration in section 4 where they provide thumb rules on the number of gauges required at each time scale.

2) Consistency in conclusions from Figures 5 and 6: Firstly, if the metric plotted along y-axis of Figure 6 is the ratio of resampled rainfall to true rainfall, then it cannot be negative. Is it defined as (resampled-truth)/truth? The y-axis label and the figure caption should be changed accordingly. While discussing Figure 5, the authors state that "At the daily scale, more than one gauge per 300 km2 would be sufficient to reach

the >20% accuracy level. Correspondingly, at the temporal scales of 1-h, 30-min, and 5-min, more than 12, 18, and 33 gauges, respectively, are needed to achieve the same level of accuracy". In lines 15-17 of Page 7, the study concludes that at least 2-5 gauges are required for reliable areal rainfall estimates with "no significant error" and "More than 5 gauges guarantee a high accuracy of areal rainfall estimates". This conclusion of 2-5 gauges is specific to Figure 6. But the manner in which it is mentioned in Page 7 overrules the conclusions reached via Figure 5 and gives an impression that it is the main conclusion as far as the number of gauges required. The differences between Figure 5 (overall NRMSE for 1000 sub-networks) and Figure 6 (inter-event variability for a particular sub-network) need to be clearly discussed in section 4 and in section 6.

3) Effect of spatial scale: It is expected that the rainfall fields are smoother at coarser spatial and temporal scales. In fact, the behaviour is quantified in the form of correlograms (section 3), where the correlation distance increased with time scale. So, the novelty in the analysis presented in section 5 should be discussed clearly?

Specific comments:

1) Page 3, line 23: The authors mention that the gauges record rainfall every five minutes. As the gauges used are tipping-bucket type, I am assuming that the authors' statement implies that the raw data from the data logger is aggregated every five minutes. Please clarify.

2) Page 4, line 15: How are zero values treated during log-transformation of the data before estimating correlograms? Furthermore, a discussion on how the readers should interpret the correlograms obtained in the log-transformed space would be useful. For example, the correlation distance of 200 km for the wet season and daily scale is based on the log-transformed data. So, I wonder how this correlation distance in log-transformed data reflects in the non-transformed space (i.e. rainfall values). On the same note, can the authors elaborate why Pearson's correlation is used over Spearman's correlation? The latter is more robust to non-normality of the data.

3) Page 6, lines 1-4: All rain rates are wrongly mentioned as mm/day. They should be in mm/h to be consistent with Figure 7. Also, are the percentiles in Figure 7 obtained from the full dataset unlike the analysis in section 4 with 71 events? If so, it should be mentioned at the beginning of the section 5. Moreover, 10% of WEGN records at 30-min scale are lower than 38 mm/h. Please change 40 to 38 in the text.

Minor comments:

1) Page 1, line 3: What is "similar approach"? Please add what it is similar to. 2) Page 5, line 15: Consider changing "reach the >20% accuracy" to "reach <20% estimation error". 3) Page 5, Section 5 heading: Consider changing "spatial scaling" to "spatial scale" as the former implies spatial scale-invariance. 4) Page 6, line 18: Same as above comment on "spatial scaling" 5) Page 8, line 8: It should be "afterward" 6) Figure 5: In the box-plot, what does the whisker indicate? Is it [minimum,maximum] or [5th,95th] percentile?

---

## Referee Comment (RC2) · Anonymous Referee #2 · 14 Nov 2018

**Assessment of spatial uncertainty of heavy local rainfall using a dense gauge network**
by O. Sungmin and U. Foelsche

**Summary:** A dense network of rain gauges in Austria is used to assess the spatial and temporal variability of rainfall. Special emphasis is given to quantifying the spatial autocorrelation structure for different aggregation time scales and assessing the accuracy of areal-rainfall estimates as a function of the number of available gauges.

Overall, this is a rather well written paper with clear objectives and useful new conclusions. The high-quality dataset of the WegenerNet allows for some nice experimental studies on the space-time variability of rain, providing unique insight into the importance of network density and sampling uncertainty. My main criticism concerns the lack of rigor in the the the statistical analyses (see below for more details) and the way the authors select the events for drawing their conclusions. There also appears to be some minor terminology issue related to the notion of the "nugget effect" (see below).

**Recommendation:** Major review

**Major Comments:**

**A. Event selection:**

**A1.** The way the events are selected in this study (based on total daily rainfall amounts) has some important consequences which are not discussed enough in the paper in my opinion. We know from other studies that at mid-latitudes and in continental climates, the rainfall events that produce the largest daily accumulations are generally more widespread and persistent than the ones responsible for small-scale extremes. As a consequence, there are plenty of heavy localized rainfall events with high peak intensity but low to moderate rainfall totals that the authors do not consider in this analysis. Conversely, there are events in the sample that do not have very high peak intensity. This is not necessarily wrong but has important consequences as it heavily influences the conclusions. This needs to be discussed more in detail given that the focus of this paper is on heavy localized rainfall.

**A2.** More generally, a table summarizing the properties of the events selected for the analysis would be helpful.

**B. Spatial correlation analysis:**

**B1.** The WEGN is a rectangle of 20x15 km which means that it favors the sampling of some particular spatial directions over others. For small distances this does not really matter as all spatial directions are sampled more or less uniformly. But as you start considering gauges separated by 15 km or more, the number of different spatial directions you can sample in your network decreases. This has important consequences when estimating a spatial autocorrelation function, especially in cases when the rainfall has a preferred direction of spatial orientation (i.e., anisotropy). The proper way to deal with this is to (a) choose an appropriate cutoff distance that limits these effects or (b) fit an anisotropic correlation model. The cutoff distance you used (going up to 25 km) is probably too large, which can result in biased model parameters. I recommend that you check this more carefully to make sure that your fitted model parameters aren't contaminated by it. Typically, I wouldn't go much further than 10-15 km in distance.

**B2.** The fact that you use a logarithmic transform means that zero rainfall values are excluded from the analysis. However, this could be a problem at small aggregation time scales where it is possible to observe zero rainfall at one gauge and positive values at the others. Please explain how you deal with these cases and more generally, how zeros are handled in your analysis.

**B3.** Please explain how you fit your exponential correlation function to the sample points. Do you use any weights? What's the objective function you are optimizing?

**B4.** The fact that you get large yearly differences in correlation patterns (especially in winter and at 5 min resolution) might also (partially) have to do with the fact that you force an exponential model to your data without actually checking if the data comply with this model. In other words, you also need to say something about how good your model is at representing the data. Some goodness of fit statistics would be helpful for this. There is no physical justification for the exponential model you impose and other parametric fits might be equally good or better in some situations.

**B5.** Figure 4 shows decorrelation ranges in the order of 200-600 km. Yet the maximum range you can observe in your network is 24 km. So my questions is: how much do you trust these large range estimates? And what's the uncertainty affecting them? Please provide some form of uncertainty analysis (e.g., confidence intervals) for your parameter estimates. This would also allow you to make a more precise statement about the trend in the shape factor on p.5, line 1.

**C. Nugget:** I do not agree with your use of the word "nugget" in this paper. The nugget is NOT the value of the zero distance correlation. It's the drop in the correlation value when you go from zero distance to d>0 (i.e., the discontinuity at zero). In other words, it's not $c_1$ but $1-c_1$. For example, when you say that the nugget is 0.73 to 0.98, actually, it's 0.02 to 0.27. The advantage of defining it this way is that you get a better interpretation in terms of sub-grid variability + measurement error. Large nugget = large differences at sub-grid scale. Please change and adapt the rest of the text to give the right meaning.

**D. Areal rainfall estimates:**

D1. The method used to sample the 1'000 possible combinations of gauge sub-networks is not very clear to me. Moreover, wouldn't there be a strong dependence on how the gauges are selected within the network (area of influence)? I mean, you only show graphs of accuracy as a function of the number of gauges. But obviously, having 4 gauges next to each other is not the same at all as having 4 equally spaced gauges covering the whole 20x15km area. I've read this part several times but couldn't really figure out the approach. Some further details about the approach would be helpful.

**Minor comments & typos:**

p.2, ll.6-8 "Although relatively high-resolution data from remotely […] cannot be fully captured at the sub-pixel scale". This sentence is not clear. Please reformulate.

p.2, l.13 "[…] intra-pixel variability of rainfall on the performance of remote sensing" A reference to the literature is needed here.

p.3, ll.3-7 "The accuracy of areal rainfall estimation is a long-standing issue […] high-resolution gauge data (e.g., Wood et al., 2000; Villarini et al., 2008;Ly et al., 2011)" This entire paragraph is out of context. It would be better to put it a few lines earlier in  the introduction, before you mention the structure of this paper.

p.3, ll.31-33: I'm not sure whether "wet" and "dry" seasons is really a good choice of terminology here. Wet and dry seasons are usually seen in the context of tropical climates and using them for Austria feels weird. What you have here is a continental climate, with most of the precipitation falling in the warmer months of the year. Warm and cold season would be much better choices.

p.4 ll.16-18: in this paragraph you start by saying that "we do not make a direct comparison with other studies". However, a few lines later you say that "the functions show a broad agreement with those from previous studies". I get what you wanted to say, but it's probably a good idea to reformulate the sentence to avoid the apparent contradiction here.

p. 6, ll.27-28: "+7% to +63% of increases in extreme rainfall intensities are observed depending on the considered spatial scale". Not clear what you mean by that. Please reformulate.

p.7, l.12 Replace "Seeing that only two operational [...]" by "Given that only two operational [...]"

p.7 l.11 "shows there to be a high dependence" English.

p.7 l.13 "[…] under normal circumstances could be inadequate for particular purposes". Too vague, please reformulate.

p.7, l.30 "statistical robust results." I don't think that you can claim this. You only have 10 years of data (which is not much for extremes) and you did not do any sensitivity analysis nor do you have any confidence intervals to prove this. Please reformulate.

p.8, l.8 "afterword" replace by "afterward"

---

## Author Comment (AC1) · 27 Jan 2019

**Reply to the Comments from Referee #1 for HESS-2018-517**

*We greatly appreciate the referee for thorough reading of the manuscript and valuable comments, which helped us to improve the manuscript significantly. In the following, we have provided an item-by-item reply to the comments.*

**Major Comments:**

1) Discussion on the required number of gauges: The study provides information on the number of gauges required to estimate areal rainfall within an estimation error (NRMSE) of 20%. Such information would be useful for many purposes including i) the design of new gauge networks, and ii) quantifying estimation errors from the existing networks. However it should be noted that it is BOTH the number and the spatial configuration of the network that determine the estimation error. The interplay between spatial configuration and number of gauges can be easily illustrated using hourly scale results in Figure 5 (bottom-left panel). The authors conclude here that 12 gauges are required at hourly scale for NRMSE to be < 20%. However, from Figure 5, it can be seen that the NRMSE at hourly scale can be brought to within 20% even with 8 gauges, although for a specific spatial configuration. So, from the perspective of gauge network design, meeting the desired error limit with the minimum number of gauges is more useful. On the other hand, with 24 gauges, the NRMSE at hourly scale will be below 20% for all 1000 spatial configurations considered in Figure 5. In short, 20% NRMSE at hourly scale can be achieved with 8-24 gauges depending on their spatial configuration. On average (i.e. not always), 12 gauges ensure <20% NRMSE. I suggest that the authors highlight the role of spatial configuration in section 4 where they provide thumb rules on the number of gauges required at each time scale.

> *=> Thank you for this suggestion. We agree with the reviewer that "the minimum number of gauges" to reach a certain level of error (NRMSE 20% in our study) will be interesting for certain purposes. Now this point is described in Section 4 along with more analysis to show the role of spatial configuration of the network in estimating areal rainfall uncertainty (Page 6 Lines 23-31). Among the 1,000 combinations for each n-gauge network, we selected "good" and "bad" combinations, 100 cases each, based on the uniformity index defined in Appendix B (it is the uniformity index we used for Fig.6) and calculated their NRMSE. As seen in Fig 5, small errors can be achieved when the gauges are regularly distributed.*

2) Consistency in conclusions from Figures 5 and 6: Firstly, if the metric plotted along y-axis of Figure 6 is the ratio of resampled rainfall to true rainfall, then it cannot be negative. Is it defined as (resampled-truth)/truth? The y-axis label and the figure caption should be changed accordingly. While discussing Figure 5, the authors state that "At the daily scale, more than one gauge per 300 km2 would be sufficient to reach the >20% accuracy level. Correspondingly, at the temporal scales of 1-h, 30-min, and 5-min, more than 12, 18, and 33 gauges, respectively, are needed to achieve the same level of accuracy". In lines 15-17 of Page 7, the study concludes that at least 2-5 gauges are required for reliable areal rainfall estimates with "no significant error" and "More than 5 gauges guarantee a high accuracy of areal rainfall estimates". This

conclusion of 2-5 gauges is specific to Figure 6. But the manner in which it is mentioned in Page 7 overrules the conclusions reached via Figure 5 and gives an impression that it is the main conclusion as far as the number of gauges required. The differences between Figure 5 (overall NRMSE for 1000 sub-networks) and Figure 6 (inter-event variability for a particular sub-network) need to be clearly discussed in section 4 and in section 6.

*=> We have corrected the Y-axis label and the figure caption (Page 19). Thanks for the correction. As the referee pointed out, we showed the overall estimation error from random possible gauge combinations using Fig.5 and examined the event-based rainfall estimation error from a "well-designed" network using Fig.6. Now this is clearly pointed out in Sect.6 (e.g., Page 8 Lines 28-32 vs Page 8 Line 33- Page 9 Line 3).*

3) Effect of spatial scale: Effect of spatial scale: It is expected that the rainfall fields are smoother at coarser spatial and temporal scales. In fact, the behaviour is quantified in the form of correlograms (section 3), where the correlation distance increased with time scale. So, the novelty in the analysis presented in section 5 should be discussed clearly?

*=> Section 5 intends to show the uncertainty in final rainfall products due to the selected spatial resolution, no matter how detailed spatial information (from many gauges) was obtained. We aim to quantify how different rainfall events will be considered extreme (Fig.8) depending on spatial and temporal resolution (e.g., extreme rainfall at 5-min scale varies from 54.2 mm to 78.2 mm as resolution increases from 0.1° to 0.01°). In addition, we aim to demonstrate how rainfall data resolution affects frequency and location information of extreme rainfall events (Fig.9) which couldn't be deduced from correlograms (Sect. 3). Please refer to Sect 5 (Page 7) where we made modification.*

**Specific Comments:**

1) Page 3, line 23: Page 3, line 23: The authors mention that the gauges record rainfall every five minutes. As the gauges used are tipping-bucket type, I am assuming that the authors' statement implies that the raw data from the data logger is aggregated every five minutes. Please clarify.

*=> yes, that is correct. We have rewritten the sentence as "Raw rain gauge data are aggregated every five minutes"; now Page 3 Line 33.*

2) Page 4, line 15: Page 4, line 15: How are zero values treated during log-transformation of the data before estimating correlograms? Furthermore, a discussion on how the readers should interpret the correlograms obtained in the log-transformed space would be useful. For example, the correlation distance of 200 km for the wet season and daily scale is based on the log-transformed data. So, I wonder how this correlation distance in logtransformed data reflects in the non-transformed space (i.e. rainfall values). On the same note, can the authors elaborate why Pearson's correlation is used over Spear-man's correlation? The latter is more robust to non-normality of the data.

*=> we used rain value+1 to include zero rainfall; log(0+1)=0. We added more details about how to calculate correlations and the sensitivity of the correlation estimation to log-transformation (from Page 4 Line 31). Interpretation of correlations from transformed data is not straightforward, however, we checked that correlation obtained from logtransformed vs non-logtransformed data yield pretty similar results (please see the figure below).*

[Figure]

*Figure 1 Correlation Distance derived from log-transformed vs non-tranformed data*

*In addition, we checked the effect of fitting a model to the observed correlation, which was questioned by Referee #2 (comment B4); it turned out that the correlation distance is more sensitive to the fitting model, rather than log-transformation, especially when the spatial scale of the observed correlation is limited to a distance of a few kilometers. This is further discussed in Section 3 (from Page 5 Line 30 & Page 9 25-28).*

*Please note that Fig.4 is updated. i) RMSE of observed vs fitted correlation is added ii) the model is fitted to correlation values over ranges < 15 km to sample data uniformly in any spatial direction (Page 4 Lines 22-26)*

*One could use Spearman's correlation, however the Pearson's r is the most commonly used rainfall correlation estimator (e.g., Krajewski et al. 2003; Ciach and Krajewski 2006; Villarini et al., 2008; Peleg et al., 2013; Tokay et al., 2014 and more\*). To our knowledge, Spearman's correlation is often used to compare rainfall distribution from different sensors (e.g. gauge vs satellite), but barely used to characterize spatial structure of rainfall; meanwhile, Svoboda et al (2015\*) used Spearman's and Kendall's correlation in addition to Pearsons'r to compare each other. Due to this reason, we would prefer using Pearson's r for consistency with other studies.*

*\* all references are listed in the manuscript*

3) Page 6, lines 1-4: All rain rates are wrongly mentioned as mm/day. They should be in mm/h to be consistent with Figure 7. Also, are the percentiles in Figure 7 obtained from the full dataset unlike the analysis in section 4 with 71 events? If so, it should be mentioned at the beginning of the section 5. Moreover, 10% of WEGN records at 30-min scale are lower than 38 mm/h. Please change 40 to 38 in the text.

*=> We apologize for the mistake. The unit should be "mm/h" and this is now corrected (Page7 Lines 13-14). Only datasets from the 71 events are used for Section 4 to 5. We added a sentence of "for the heavy rainfall events" at the beginning of Sect. 5 to make it clear (Page 7 Line 10) and also mentioned it in Appendix A (Page 10). We also changed the rain intensity 40 mm/h to 38 mm/h (Page 7 Line 14).*

**Minor comments:**

1) Page 1, line 3: What is "similar approach"? Please add what it is similar to.
   *=> the sentence is rewritten: "a similar approach, i.e., employing dense gauge networks to catchment-scale areas" (Page 1 Line 3)*

2) Page 5, line 15: Consider changing "reach the >20% accuracy" to "reach <20% estimation error".
   *=> we changed the sentence to "reach <20% estimation error" (Page 6 Line 19)*

3) Page 5, Section 5 heading Consider changing "spatial scaling" to "spatial scale" as the former implies spatial scale-invariance.
   *=> the heading is now "spatial aggregation" (now Page 7)*

4) Page 6, line 18: Same as above comment on "spatial scaling"
   *=> changed to "spatial aggregation" (Page 7 Line 31)*

5) Page 8, line 8: It should be "afterward"
   *=> corrected (Page 10 Line 11)*

6) Figure 5: In the box-plot, what does the whisker indicate? Is it [minimum,maximum] or [5th,95th] percentile?
   *=> the whisker indicates $10^{th}$-$90^{th}$ percentiles. The figure caption is updated.*

[revised manuscript text omitted]

---

## Author Comment (AC2) · 27 Jan 2019

**Reply to the Comments by Referee #2 for HESS-2018-517**

*We greatly appreciate the referee's insightful comments along with many helpful suggestions, which helped us to improve the quality of the manuscript. We have faithfully revised the manuscript following the referee's comments and suggestions. Please find below our item-by-item replies to the referee's comments.*

**Major Comments:**

*A. Event selection:*

*A1. The way the events are selected in this study (based on total daily rainfall amounts) has some important consequences which are not discussed enough in the paper in my opinion. We know from other studies that at mid-latitudes and in continental climates, the rainfall events that produce the largest daily accumulations are generally more widespread and persistent than the ones responsible for small-scale extremes. As a consequence, there are plenty of heavy localized rainfall events with high peak intensity but low to moderate rainfall totals that the authors do not consider in this analysis. Conversely, there are events in the sample that do not have very high peak intensity. This is not necessarily wrong but has important consequences as it heavily influences the conclusions. This needs to be discussed more in detail given that the focus of this paper is on heavy localized rainfall.*

> *=> This study aims to provide spatial uncertainty information of heavy rainfall events in a general sense, not only targeting moderated/localized rainfall events (Appendix A, Page 10), given that, i) it is very common to define heavy or extreme rainfall events using a certain threshold value (e.g., Zhang et al., 2001; Zhai et al., 2005; Villarini 2012; Salack et al., 2018, etc.) and, ii) in many studies, gauge-based data are used for, e.g., remote sensing data validation or runoff modeling, with a lack of consideration about the rain type.*

> *Nonetheless, we agree with the referee that the way of selecting rainfall events can strongly influence quantitative results (Page 8 Lines 20-22, Page 9 Line22). This remains as a limitation of our study and open question for further study. To avoid any confusion, we have changed the title of manuscript to: "Assessment of spatial uncertainty of heavy rainfall at catchment scale using a dense gauge network"*

*A2. More generally, a table summarizing the properties of the events selected for the analysis would be helpful.*

> *=> Thanks for the suggestion. We added a table in Appendix A (Page 24).*

*B. Spatial correlation analysis:*

B1. *The WEGN is a rectangle of 20x15 km which means that it favors the sampling of some particular spatial directions over others. For small distances this does not really matter as all spatial directions are sampled more or less uniformly. But as you start considering gauges separated by 15 km or more, the number of different spatial directions you can sample in your network decreases. This has important consequences when estimating a spatial autocorrelation function, especially in cases when the rainfall has a preferred direction of spatial orientation (i.e., anisotropy). The proper way to deal with this is to (a) choose an appropriate cutoff distance that limits these effects or (b) fit an anisotropic correlation model. The cutoff distance you used (going up to 25 km) is probably too large, which can result in biased model parameters. I recommend that you check this more carefully to make sure that your fitted model parameters aren't contaminated by it. Typically, I wouldn't go much further than 10-15 km in distance.*

> *=> Thanks for the comment. It is true that we have a smaller number of data samples from North-South direction rainfall events at >15 km. We have re-calculated fitting models using correlation values up to and including 15 km (Page 4 Lines 24-26) and updated the figures 3 and 4 accordingly. The new figure 4 shows clearer difference in Shape factor among seasons (compared to the previous version).*

B2. *The fact that you use a logarithmic transform means that zero rainfall values are excluded from the analysis. However, this could be a problem at small aggregation time scales where it is possible to observe zero rainfall at one gauge and positive values at the others. Please explain how you deal with these cases and more generally, how zeros are handled in your analysis.*

> *=> We included zero rainfall values by adding +1 to the rainfall data; $log(0 +1) = 0$. This is now explained with more details in Sect.4 (Page 4/Line 30 ~ Page5/Line3), where we describe the correlation calculation method.*

B3. *Please explain how you fit your exponential correlation function to the sample points. Do you use any weights? What's the objective function you are optimizing?*

> *=> We chose the parameters of fitting functions in order to minimize the sum of the squared residuals. This information is added at Page 4, Lines 29/30.*

B4. *The fact that you get large yearly differences in correlation patterns (especially in winter and at 5 min resolution) might also (partially) have to do with the fact that you force an exponential model to your data without actually checking if the data comply with this model. In other words, you also need to say something about how good your model is at representing the data. Some goodness of fit statistics would be helpful for this. There is no physical justification for the exponential model you impose and other parametric fits might be equally good or better in some situations.*

*=> V.Svoboda et al (2014) well summarized the fitting models that are commonly used to represent spatial rainfall correlation functions; most studies adopted 3-parameters functions, while some studies used 2-parametes functions. We had compared several different forms of 2- and 3-parameters models including the models listed in V.Svoboda et al (2014) and found that the selected 3-parameters model works well across all temporal scales in terms of RMSEs of original correlations vs fitted correlations (Figure 4 is updated).*

*The figure below shows an example of comparing the RMSEs of the fitting model used in this study (left) and RMSEs of 2-parameter models tested (middle and right). In any cases, we observe yearly differences in correlation patterns (especially in winter at 5-min, not shown). However, it turned out that Correlation Distance can be significantly affected by the selection of the fitting model. This point will be further discussed in the following comment B5.*

[Figure]

B5. *Figure 4 shows decorrelation ranges in the order of 200-600 km. Yet the maximum range you can observe in your network is 24 km. So my questions is: how much do you trust these large range estimates? And what's the uncertainty affecting them? Please provide some form of uncertainty analysis (e.g., confidence intervals) for your parameter estimates. This would also allow you to make a more precise statement about the trend in the shape factor on p.5, line 1.*

*=> we have to admit that the correlation distance estimated from observation that are only available in a distance of a few kilometers (>6-hr, especially for cold season) is highly uncertain; i.e., the fitting model plays a dominant role in estimating $c_2$ values. However, no matter what fitting model is used for obtaining the parameter values, the general behaviors of parameter like their difference among seasons, among time scales remain the same. These points are now addressed in Sec.3 (from Page 5 Line 30 & Page 9 Lines 25-28)*

*C. Nugget: I do not agree with your use of the word "nugget" in this paper. The nugget is NOT the value of the zero distance correlation. It's the drop in the correlation value when you go from zero distance to d>0 (i.e., the discontinuity at zero). In other words, it's not c1 but 1-c1. For example, when you say that the nugget is 0.73 to 0.98, actually, it's 0.02 to 0.27. The advantage of defining it this way is that you get a better interpretation in terms of sub-grid variability + measurement error. Large nugget = large differences at sub-grid scale. Please change and adapt the rest of the text to give the right meaning.*

> *=> We agree with the referee that "1-c1" provides a better interpretation (as does in rainfall semivariogram), however, since most studies refer to "c1" as "Nugget" (e.g., Villarini et al., 2008; Peleg et al., 2013; Tokay et al., 2014 and more; all are listed in the manuscript), we would prefer to follow convention for this parameter. Please note that we replaced "nugget" by "nugget effect" (Page 4 Line 28), "higher zero-distance correlation" by "higher c1 (smaller microscale variations)" (Page5 Line10) and "The nugget implies measurement errors [...]" by "lower c1 values [...], meaning larger measurement errors [...]" (Page5 Line17).*

*D. Areal rainfall estimates:*

*D1. The method used to sample the 1'000 possible combinations of gauge sub-networks is not very clear to me. Moreover, wouldn't there be a strong dependence on how the gauges are selected within the network (area of influence)? I mean, you only show graphs of accuracy as a function of the number of gauges. But obviously, having 4 gauges next to each other is not the same at all as having 4 equally spaced gauges covering the whole 20x15km area. I've read this part several times but couldn't really figure out the approach. Some further details about the approach would be helpful.*

> *=> The possible range of errors in areal rainfall estimates with a fixed number of gauges (often without a consideration of gauge configuration) has been studied, e.g., to see the reliability of gauge-based ground reference for satellite data evaluation (Tian et al., 2018) or to see the influence of rain gauge density on hydrological model performance (Xu et al., 2013; references are listed in the manuscript). In this context, we provide the average and the spread of areal rainfall uncertainty as a function of gauge number, using 1,000 random combinations. Please note that we checked that the number of 1,000 is enough to represent variation of the overall estimation error; i.e., box plots are not significantly changed no matter which 1,000 combinations are selected.*

> *As the referee pointed out, we didn't discuss the impact of gauge configuration. To address this point, using the area of influence (the index defined in Appendix B), we selected the best and the worst configurations (100 cases, respectively) out of 1,000 combinations for each n-gauge network and calculated the error of the best and worst configurations; the results show that gauge configuration strongly determines the accuracy of areal rainfall estimates and we have addressed this point adequately in the revised manuscript, which appears in Page 6 Lines 23-27 with Fig.05-a.*

*In addition, following the suggestion of the Referee #1, we also demonstrated the minimum number of gauges to meet the desired error limit, which would be interesting from the perspective of gauge network design; please see Page 6 Lines 27-31 and Fig.05-b.*

**Minor comments & typos:**

*We thank the referee for the careful reading of our manuscript.*

p.2, ll.6-8 "Although relatively high-resolution data from remotely [...] cannot be fully captured at the sub-pixel scale". This sentence is not clear. Please reformulate.

> *=> rewritten as "Gridded rainfall data from remotely sensed observations are nowadays available at high spatial resolutions. While those data sets are good alternatives to address a number of the issues relating to the scarcity of gauges, rainfall variability at sub-pixel scales can still not be fully resolved." (Page 2 Lines 7-11)*

p.2, l.13 "[...] intra-pixel variability of rainfall on the performance of remote sensing" A reference to the literature is needed here.

> *=> references are added (Page 2 Line 17).*

p.3, ll.3-7 "The accuracy of areal rainfall estimation is a long-standing issue [...] high-resolution gauge data (e.g., Wood et al., 2000; Villarini et al., 2008;Ly et al., 2011)" This entire paragraph is out of context. It would be better to put it a few lines earlier in the introduction, before you mention the structure of this paper.

> *=> the paragraph explains the motivation of Sect.4. We would therefore prefer to leave the paragraph as it is. A sentence of "We followed the latter approach using the WEGN rainfall data." has been added to Page 3 Line 16.*

p.3, ll.31-33: I'm not sure whether "wet" and "dry" seasons is really a good choice of terminology here. Wet and dry seasons are usually seen in the context of tropical climates and using them for Austria feels weird. What you have here is a continental climate, with most of the precipitation falling in the warmer months of the year. Warm and cold season would be much better choices.

> *=> Thank you for this comment. We have decided to follow the referee's suggestion; now "wet" and "dry" are changed to "warm" and "cold" throughout the manuscript.*

*p.4 ll.16-18: in this paragraph you start by saying that "we do not make a direct comparison with other studies". However, a few lines later you say that "the functions show a broad agreement with those from previous studies". I get what you wanted to say, but it's probably a good idea to reformulate the sentence to avoid the apparent contradiction here.*

> *=> this sentence is modified as "[...] We therefore do not make a direct comparison of correlation values with those from other studies, yet we still observe that the behaviors of the correlation decay found in this study are in broad agreement with rainfall spatial correlation structure reported in the aforementioned studies (Page 5 /Lines 6-8).*

*p. 6, ll.27-28: "+7% to +63% of increases in extreme rainfall intensities are observed depending on the considered spatial scale". Not clear what you mean by that. Please reformulate.*

> *=> the sentence is rewritten as "[...] The 10-year rainfall maximum appears to be 68.4 mm/day at HR10, but 104.4mm/day at HR01; the maximum record over the entire WEGN area is 64.1 mm/day, so the ratio of the site-to-areal extreme rainfall ranges from 1.07 to 1.63 depending on the considered spatial scale." (Page 8 Lines 7-10)*

*p.7, l.12 Replace "Seeing that only two operational [...]" by "Given that only two operational [...]"*

> *=> replaced (Page 8 Line 29)*

*p.7 l.11 "shows there to be a high dependence" English.*

> *=> "shows a clear dependence" (Page 8 Line 27)*

*p.7 l.13 "[...] under normal circumstances could be inadequate for particular purposes". Too vague, please reformulate.*

> *=> modified as "[...] the insufficient gauge density may hamper the use of the station data to construct spatial rainfall fields in the region, especially at sub-daily scales." (Page 8 Lines 31-32)*

*p.7, l.30 "statistical robust results." I don't think that you can claim this. You only have 10 years of data (which is not much for extremes) and you did not do any sensitivity analysis nor do you have any confidence intervals to prove this. Please reformulate.*

*=> corrected (Page 9 Line 20): "long-term records, which permits to exclusively focus on heavy rain events"*

*p.8, l.8 "afterword" replace by "afterward"*

*=> corrected (Page 10 Line11). Thanks.*

[revised manuscript text omitted]

---

## Referee Report (RR1)

**Assessment of spatial uncertainty of heavy rainfall at catchment scale using a dense gauge network**

by Sungmin O and Ulbrich Foelsche

This is the second time I review this paper. Most of my previous comments were taken into account and the authors have made substantial changes to the original manuscript. As a result, the quality has improved. That being said, there are still a few important issues that remain to be addressed before publication can be envisaged.

**Major Comments:**

**1. Usefulness and novelty:**

While the WegenerNet is an amazing and unique source of data, the paper itself is rather dull and empty, providing very few ideas and findings worth publishing. The conclusions can be summarized as follows:

- The spatial variability of rainfall varies with years, seasons and events (no big surprise there). The spatial autocorrelation structure is anisotropic and approximately decreases exponentially with distance (which has been known for a long time already).

- The small-scale variability during the cold season is higher while the decorrelation range is longer, in agreement with previous findings.

- The uncertainty in areal-rainfall estimates depends on the number of gauges available (obviously!). High-resolution gridded data provide more reliable information about extremes (also obvious).

Many of the numbers are specific to the WegenerNet and of little usefulness to others. The only new idea in my opinion is the power-law model for predicting the number of gauges needed to ensure the uncertainty affecting mean areal rainfall estimates is below a given threshold. But this part of the paper is not detailed enough and the model has not been properly validated (see comment 3). Otherwise, the paper remains very descriptive, with only a single equation in it. There's nothing wrong with it but I just don't think that it will be useful. A more useful approach in my opinion would have been to use the WegenerNet data to formulate predictive models for assessing uncertainties and validating findings on other networks or situations.

**2. Too many inconsistencies:**

The paper contains many inconsistencies. Often, the text says A while the figures show B.

- in the autocorrelation function, the text says that the largest considered distance is 15 km. Yet Figure 3 shows distances up to 20 km and more. This was already an issue in the previous version and has not been addressed properly.

- On page 4, the text says that a data transformation is necessary to deal with zeros and make the data more Gaussian distributed. Yet Figure 3 shows the autocorrelation without data transformation (or at least I assume, it's not 100% clear). If indeed the values change after transformation, why don't you show the ones matching the text?

- For the exponential model, the text mentions that another two-parameter model was tried with very similar results. Yet somehow the authors decided to go for the three parameter model anyway. The explanations to justify this choice are not convincing at all and proper validation is required to motivate this choice. Indeed, one of the parameters seems to be almost constant across scales, which means its probably not useful.

- Page 8, line 24: the text says that the study has confirmed that the WEGN provides very accurate areal-precipitation estimates. But actually, there is no evidence for this in the paper. Only comparisons where the average of the 150 stations is assumed to represent the truth.

- Page 8, lines 32-33: "More than 10 gauges guarantee that we can obtain constant results regardless of the number of gauges." This is not true! The only evidence you show is that the average error will be lower than 20%.

**3. Validity of power-law model:**
The power law model proposed in Figure 6 is potentially interesting as one could imagine situations in which people need to estimate the required gauge density for achieving a certain accuracy. However, it should be pointed out that (a) it would be better to formulate it in terms of gauge density (#gauges/km2) and (b) that such a model needs to be properly validated/assessed. Right now, it is just given without any further evaluation or critical discussion. One big question is how well does the power-law model generalize to other cases. For example, what if I work with a catchment of 100 km2, that is, 3x times smaller than the area of the WegenerNet? Does this mean that I can divide the number of gauges by 3? I assume not since the autocorrelation varies faster over the first 10 km compared with the 10-20 km range. There is a lot of potential here for formulating a model that can be used by others. But this requires additional work.

**4 Data transformation:**
On page 4 line 26 it is said that a transformation $x \rightarrow \log(x+1)$ is applied to the data to avoid issues due to zeros. However, there are crucial details missing. For example, you need to be aware of the fact that due to the non-linearity of the log transform, the results of the correlation analysis will depend on the units in which x is expressed (e.g., using mm/h or mm). This is a common statistical fallacy and needs to be addressed more carefully. Moreover, it is still unclear to me how zeros are actually handled by the authors. Do you take all time steps (including the ones where all gauges record zeros) or only a subset? I asked for more details about this in the previous round of review but there hasn't been much progress on this aspect. Same for the anisotropy maps in Figure 5. You need to specify how zeros were handled and whether the maps show the values obtained after data transformation or before.

**5 Selection of events:**
It's not 100% clear from the text how the events in Section 4 were selected. The corresponding sentence on Page 6, lines-6-7 is not well formulated and open to interpretation. Please revise. Also, the selection seems to be done based on total rainfall accumulation, which tends to favor certain types of events (i.e., persistent and widespread), potentially providing biased results. Please comment on this in the paper.

**Minor issues:**

- Page 1, abstract: "these dense networks are only available at sub-pixel scales and over short periods of time". Too vague. Please explicitly state what pixels we are talking about. There are large differences in weather radar products and some of the latest X-band products have resolutions as high as as 30 meters.

- In Equation 1, I suggest to replace c1 by 1-c1. This would make the interpretation easier, making c1 the drop in autocorrelation (= the nugget) instead of the intercept. Higher c1 would then mean higher small-scale variability which makes more sense in my opinion.

- Figure 4: the units are missing

- Figure 4, the RMSE values (<0.01) you provide are misleading. They give the impression to the reader that the exponential fit is very good whereas the RMSE in the plot is the one that you

calculated by taking the average value of the yearly-averaged autocorrelation values. However, individual fits (for a given year or a season) are not that good and the spread remains large, as shown in Figure 3.

- Figure 4: Most decorrelation distances shown here are much larger than the maximum observable size of the network (20 km). So how much do you trust these values? The text provides a small warning and a reference to a paper but I believe this warrants more attention than that. Having worked on spatial structures myself, I know that such large range estimates are often the result of bad fits or the choice of the model rather than physical. In any case, I think it's illusory to think that you can infer a 200 km range from data extending over 20 km. A better approach would be to limit the range of scales.

- There seems to be a confusion between the e-folding distance (which is a distance and should be in units of km) and the autocorrelation corresponding to the e-folding distance (which is unitless).

- Figure 9: it's almost impossible to see any difference in color here. Please adapt the scales.

- Section 4. It would be more useful to talk about gauges/km2 to avoid introducing too much dependence on the total area. In fact, the entire discussion should be centered around gauge density rather than the number of gauges. Also, other thresholds than 20% should be considered, as this is rather large in my opinion and is more characteristic of the accuracy one would like to achieve at the point scale.

- Page 5, lines 10-14: You mainly attribute the higher small-scale variability to solid or mixed precipitation here. Another explanation is that winter-type events are more heterogeneous and spatially disorganized than convective cases. Moreover, since they are lower intensity, the uncertainty affecting the measurements of two neighboring stations plays a bigger role. Bottom line: there are more explanations that can be given here and it's not clear from the evidence that you present that the higher small-scale variability is indeed attributed to snow and ice. Please rephrase.

**Typos:**
- Page 2, line 7: "On the contrary, gridded rainfall data are nowadays available..." The expression "On the contrary" does not appear adequate here.

- Page 2, line 34, "in order to contribute to the effort better and more broadly assessing the uncertainty". Bad English, please rephrase.

- Page 7, line 25: "decreases" instead of "deceases"

- Figure 3: there is a typo in the caption (separation)

- Figure 5: there is a typo in the caption (north-south)

---

## Author Response (AR2)

**Reply to the Comments from Referee #1 for HESS-2018-517**

This is the second time I review this paper. Most of my previous comments were taken into account and the authors have made substantial changes to the original manuscript. As a result, the quality has improved. That being said, there are still a few important issues that remain to be addressed before publication can be envisaged.

*We appreciate the time and effort the reviewer put into the review. In the following, we have provided an item-by-item reply to the comments. Please note that lines and pages mentioned here are based on the marked-up manuscript.*

**Major Comments:**

**1. Usefulness and novelty:**

While the WegenerNet is an amazing and unique source of data, the paper itself is rather dull and empty, providing very few ideas and findings worth publishing. The conclusions can be summarized as follows:

- The spatial variability of rainfall varies with years, seasons and events (no big surprise there). The spatial autocorrelation structure is anisotropic and approximately decreases exponentially with distance (which has been known for a long time already).

- The small-scale variability during the cold season is higher while the decorrelation range is longer, in agreement with previous findings.

- The uncertainty in areal-rainfall estimates depends on the number of gauges available (obviously!). High-resolution gridded data provide more reliable information about extremes (also obvious).

Many of the numbers are specific to the WegenerNet and of little usefulness to others. The only new idea in my opinion is the power-law model for predicting the number of gauges needed to ensure the uncertainty affecting mean areal rainfall estimates is below a given threshold. But this part of the paper is not detailed enough and the model has not been properly validated (see comment 3). Otherwise, the paper remains very descriptive, with only a single equation in it. There's nothing wrong with it but I just don't think that it will be useful. A more useful approach in my opinion would have been to use the WegenerNet data to formulate predictive models for assessing uncertainties and validating findings on other networks or situations.

*=> As pointed out by Reviewer, the main limitation of studies using a local network such as WEGN is that it is hard to generalize the studies' findings and results to other regions. However, such local-scale studies can support a comprehensive understanding of rainfall processes, not just by delivering results found from different climate/rainfall regimes, but also by showing the potential use of the network for future (collaborative) research.*

*In addition, WEGN data have been widely used by science communities and the need for the data is increasing. Therefore, this study is also motivated by the practical needs of on-*

*going/future remote sensing applications and modeling studies.*

*It might be true that our general findings regarding the spatial variability of rainfall are not totally new as pointed out by the reviewer, however we believe that the quantitative information derived from WEGN is valuable not only for local science communities (who are using WEGN data directly) but also for hydrology communities (who want to compare rainfall characteristics before applying our results, e.g., the power-law model, to their own regions).*

*The knowledge about the approximately exponential autocorrelation decrease with distance is based on a limited number of studies in different geographic regions and climate regimes; very few of them have been performed with a comparable spatial resolution. We believe that it is also important to confirm previous knowledge.*

*It may not be surprising that the uncertainty in areal-rainfall estimates depends on the number of gauges, but we provide detailed information on \*how\* it depends on the number auf gauges.*

*We agree with the reviewer that the conclusions did not fully reflect the main results of the paper and added some quantitative information (also to the abstract).*

*Please refer to Comment 3 for more details on the power-law model.*

**2. Too many inconsistencies:**

The paper contains many inconsistencies. Often, the text says A while the figures show B.

- in the autocorrelation function, the text says that the largest considered distance is 15 km. Yet Figure 3 shows distances up to 20 km and more. This was already an issue in the previous version and has not been addressed properly.

> *=> The correlation model **is built** using data of separation distance <=15 km and, for the Fig. 3, the model **is used to present** the correlation function till ~23 km. We clarify this in the caption of Fig. 3.*

- On page 4, the text says that a data transformation is necessary to deal with zeros and make the data more Gaussian distributed. Yet Figure 3 shows the autocorrelation without data transformation (or at least I assume, it's not 100% clear). If indeed the values change after transformation, why don't you show the ones matching the text?

> *=> Data after the log-transformation are used for Figure 3. We added a sentence to make it clear. 6-7 lines, Page5.*

- For the exponential model, the text mentions that another two-parameter model was tried with very similar results. Yet somehow the authors decided to go for the three parameter model

anyway. The explanations to justify this choice are not convincing at all and proper validation is required to motivate this choice. Indeed, **one of the parameters seems to be almost constant across scales**, which means its probably not useful.

> *=> The three-parameter model is selected because the model shows the smallest RMSE (fitting errors) among the tested models (lines 34-35, page5). We think "the one of parameters seems to be almost constant" means Figure4-d (RMSE); the model shows almost constant errors across scales and that is another reason why we chose the model.*

- Page 8, line 24: the text says that the study has confirmed that the WEGN provides very accurate areal-precipitation estimates. But actually, there is no evidence for this in the paper. Only comparisons where the average of the 150 stations is assumed to represent the truth.

> *=> Figures 6 and 7 show the convergence of error in areal rainfall estimation with increasing gauge number. This allows to infer that the WEGN 150 gauge provides very accurate areal-precipitation estimates. We rephrased the sentence, now at Lines 22-23, Page 9.*

- Page 8, lines 32-33: "More than 10 gauges guarantee that we can obtain constant results regardless of the number of gauges." This is not true! The only evidence you show is that the average error will be lower than 20%.

> *=> We wanted to say that the magnitude and spread of errors are not significantly changed with >=10 gauges. The sentence is rephrased (list line, Page 9 to first line, Page 10)*

**3. Validity of power-law model:**

The power law model proposed in Figure 6 is potentially interesting as one could imagine situations in which people need to estimate the required gauge density for achieving a certain accuracy. However, it should be pointed out that (a) it would be better to formulate it in terms of gauge density (#gauges/km2) and (b) that such a model needs to be properly validated/assessed. Right now, it is just given without any further evaluation or critical discussion. One big question is how well does the power-law model generalize to other cases. For example, what if I work with a catchment of 100 km2, that is, 3x times smaller than the area of the WegenerNet? Does this mean that I can divide the number of gauges by 3? I assume not since the autocorrelation varies faster over the first 10 km compared with the 10-20 km range. There is a lot of potential here for formulating a model that can be used by others. But this requires additional work.

> *=> Thanks for the comment and the suggestion. We tested the plot additionally with areas of 50 $km^2$, 100 $km^2$ and 150 $km^2$ (about 1/6, 1/3 and 1/2 of the WEGN area, respectively) and added the results to the Figure 6-b and -c. For any cases, we still observe the power-low relation between the gauge numbers to reach a certain accuracy*

*level vs time resolution. But, as the reviewer has assumed, the required gauge numbers do not linearly decrease as the considered network area decreases. This is discussed at Lines 18-28 Page 7. Please note that the gauge density information is added to the Figure 6.*

**4 Data transformation:**

On page 4 line 26 it is said that a transformation $x \rightarrow \log(x+1)$ is applied to the data to avoid issues due to zeros. However, there are crucial details missing. For example, you need to be aware of the fact that due to the non-linearity of the log transform, the results of the correlation analysis will depend on the units in which x is expressed (e.g., using mm/h or mm). This is a common statistical fallacy and needs to be addressed more carefully. Moreover, it is still unclear to me how zeros are actually handled by the authors. Do you take all time steps (including the ones where all gauges record zeros) or only a subset? I asked for more details about this in the previous round of review but there hasn't been much progress on this aspect. Same for the anisotropy maps in Figure 5. You need to specify how zeros were handled and whether the maps show the values obtained after data transformation or before.

> *=> We used mm for the units, not mm/h (i.e., no unit conversion); indeed, log-transformation after the unit conversion will significantly alter the correlation of the raw data. Data pairs, when both are no-rain, are excluded for calculation. We added this information at Line27, Page4 & Line 4, Page 5. Please note that the same data (i.e., no unit conversion, exclusion of zero-rainfall pairs, and log-transformed) are used for Figs 3-5.*

**5 Selection of events:**

It's not 100% clear from the text how the events in Section 4 were selected. The corresponding sentence on Page 6, lines-6-7 is not well formulated and open to interpretation. Please revise. Also, the selection seems to be done based on total rainfall accumulation, which tends to favor certain types of events (i.e., persistent and widespread), potentially providing biased results. Please comment on this in the paper.

> *=> The sentence is rephrased (Lines 25-26, Page 6). The heavy event is selected based on total rainfall accumulation, without a consideration of rain type. We agree with the reviewer that the way of selecting rainfall events can heavily influence quantitative results, however, we intended to define the heavy events in a general and common way – using a fixed threshold.*
>
> *We should note that the selected rainfall type is rather mixed, not biased to persistent and widespread events. This is inferred from our visual-inspection on rainfall time series (Lines 27-28, Page 6).*

**Minor issues:**

- Page 1, abstract: "these dense networks are only available at sub-pixel scales and over short

periods of time". Too vague. Please explicitly state what pixels we are talking about. There are large differences in weather radar products and some of the latest X-band products have resolutions as high as as 30 meters.

> *=> The sentence is modified (Lines 2-3, Page 1). We wanted to say that the area of other dense gauge networks is limited, i.e., smaller than remote-sensing pixel sizes in a general sense (Lines 24-25, Page 2), while the WEGN covers an area of 300 km².*

- In Equation 1, I suggest to replace c1 by 1-c1. This would make the interpretation easier, making c1 the drop in autocorrelation (= the nugget) instead of the intercept. Higher c1 would then mean higher small-scale variability which makes more sense in my opinion.

> *=> Given that many studies of rainfall correlation refer to "c1" as "nugget parameter" or "nugget effect", we would prefer to follow the convention for this parameter (see, e.g., Ciach and Krajewski, 1999, 2006; Villarini et al., 2008; Peleg et al., 2013; Svoboda et al, 2014; Tokay et al., 2014;all listed in manuscript). This is somehow in contrast to the studies of rainfall semivariograms where higher "nugget" means higher small-scale variability.*

- Figure 4: the units are missing
> *=>We added the units. Thanks for the comment.*

- Figure 4, the RMSE values (<0.01) you provide are misleading. They give the impression to the reader that the exponential fit is very good whereas the RMSE in the plot is the one that you calculated by taking the average value of the yearly-averaged autocorrelation values. However, individual fits (for a given year or a season) are not that good and the spread remains large, as shown in Figure 3.

> *=> The spread of RMSE between individual fits for a given year or a season is not that large as shown in the figure below. This is because we calculated correlation average for each distance bin first (Line 28, Page 4) and then fitted the model to the averaged correlation. This might lead to the "too-good" fitting, however, does not affect general behaviors of the parameters described in Fig4. We agree that this can be misunderstood to readers so we clarify this point at lines 3-4, page 6.*

[Figure]

*Figure 1the fitting error of the models for yearly-averaged correlation (dots) and for each year (Xs).*

- Figure 4: Most decorrelation distances shown here are much larger than the maximum observable size of the network (20 km). So how much do you trust these values? The text provides a small warning and a reference to a paper but I believe this warrants more attention than that. Having worked on spatial structures myself, I know that such large range estimates are often the result of bad fits or the choice of the model rather than physical. In any case, I think it's illusory to think that you can infer a 200 km range from data extending over 20 km. A better approach would be to limit the range of scales.

> => *we agree with the reviewer and added the 'warning sign' in the Fig.4 and modified the sentences at Lines 4-6, Page 6.*

- There seems to be a confusion between the e-folding distance (which is a distance and should be in units of km) and the autocorrelation corresponding to the e-folding distance (which is unitless).

> => *Thanks for pointing this out; we added the units to the former (Figure. 4) and revised the related sentence (Line 14, Page 6)*

- Figure 9: it's almost impossible to see any difference in color here. Please adapt the scales.

> => *the figure is updated; we added another color-scale for the second plot.*

- Section 4. It would be more useful to talk about gauges/km2 to avoid introducing too much dependence on the total area. In fact, the entire discussion should be centered around gauge density rather than the number of gauges. Also, other thresholds than 20% should be considered, as this is rather large in my opinion and is more characteristic of the accuracy one would like to achieve at the point scale.

> => *We added density information to inset-plots of Fig. 6-b and 6-c. Thanks for the*

*suggestion. The value of 20% is adopted from the well-known study of Villarini et al., 2008; we believe that readers can infer the results for other thresholds from Fig. 6-(a).*

- Page 5, lines 10-14: You mainly attribute the higher small-scale variability to solid or mixed precipitation here. Another explanation is that winter-type events are more heterogeneous and spatially disorganized than convective cases. Moreover, since they are lower intensity, the uncertainty affecting the measurements of two neighboring stations plays a bigger role. Bottom line: there are more explanations that can be given here and it's not clear from the evidence that you present that the higher small-scale variability is indeed attributed to snow and ice. Please rephrase.

> *=> Thanks for the comments. The sentence is rephrased; lines 20-24 Page 5.*

**Typos:**

- Page 2, line 7: "On the contrary, gridded rainfall data are nowadays available..." The expression "On the contrary" does not appear adequate here.

> *=> fixed.*

- Page 2, line 34, "in order to contribute to the effort better and more broadly assessing the uncertainty". Bad English, please rephrase.

> *=> rephrased.*

- Page 7, line 25: "decreases" instead of "deceases"  - Figure 3: there is a typo in the caption (separation) - Figure 5: there is a typo in the caption (north-south)

> *=> All fixed. We apologize for the typos.*

[revised manuscript text omitted]

---

## Author Response (AR3)

**Reply to the Comments for HESS-2018-517**

*Dear Nadav Peleg,*

*We thank you for your careful reading of the manuscript and helpful comments. We have made revisions according to your comments and suggestions, as described below. Please note that pages and lines used here are based on the marked-up manuscript.*

**Main comments**

[…] To keep the conclusions concise, please consider moving most of the discussion from Section 6 to Sections 3 to 5 and eliminating unnecessary repetitions.

> *=> We moved parts of Section 6 to Sections 3-5 and removed the repetitions.*

**Specific comments**

[1 19-20] I am not so sure that the results can be claimed to be general. Consider withdrawing/rephrasing this sentence. In addition, it is not supported in the text – there is no discussion about the generalization of the results beyond this sentence.

> *=> We modified the sentence. Page 1 Lines 20-22.*

[4 2] What happened after 2016?

> *=> WEGN is still in operation. The data used in this study (2007-2016) have been bias-corrected by O et al (2017). During 2016/17, most of rain-gauge sensors were replaced by 'better' sensors; no further bias analysis has been conducted for those new datasets (but all the data are quality-controlled by the WEGN data processing system). We added more information at Page 4, Lines 4-6.*

[4 4-5] Some further (basic) information is needed here. The most important - how did you treat an error measurement for a given rain gauge? Is it simply given a NAN value?

> *=> While O et al (2017) derived correction factors to mitigate systematic biases in rainfall data, data errors are managed by the WEGN quality-control processing (Kirchengast et al, 2014); e.g., errors or NAN values are filled using spatiotemporal interpolation.*

[4 7] What was the value of the power parameter? At which time scale did you interpolate? Please provide further information.

> *=> The gridded data are constructed at the native temporal resolution (i.e., 5-min), using the power parameter of 2. The sub-daily to daily data used in this study are obtained by accumulating the 5 data. This is described in Page 4, Lines 9-10.*

[4 19] Genoa Lows. Please add a reference.

> *=> added a reference. Page 4 Line 22.*

[6 25] Please present Appendix A and B as Supplementary Material, i.e. both should be moved to a separate file (not part of the main text).

> *=> Thanks for the suggestion. We moved Appendixes to Supplementary Materials.*

[8 9-10] Also see https://doi.org/10.1016/j.jhydrol.2016.05.033 Maybe of relevant.

> *=> Thanks for the suggestion.*

[8 16] Usually Fig, 9 appear before Fig. 10... Consider revising the order of the figures.

> => We revised the order of figures.

[8 24-25] A reference is needed.

> *=> added. Page 8, Line 32*

[8 27-31] It will be much more interesting to see this distribution at sub-daily scale, e.g. at hourly temporal resolution.

> *=> As we selected heavy rainfall "days" for Fig. 10-11 (now Fig. 9-10), we would prefer to keep the figures as they are. Please note that the site-to-areal extreme rainfall information (maximum values) at sub-daily scales are still available from Fig. 11.*

[9 2-5] Should be in the introduction section.

> *=> Removed. Reference merged with Page 2, Line 14.*

[9 15-17] Repetitive.
[9 23-25] Repetitive.
[10 1-3] Repetitive.
[10 6-7] Why repeating the results?

> *=> We moved parts of Sect. 6 to Sects 3-5 to keep conclusions concisely. Thanks for the suggestion.*

[revised manuscript text omitted]